1    Totten Ice Shelf history over the past century interpreted from satellite imagery

Bertie W.J. Miles[1], Tian Li[2], Robert G. Bingham[1]

[1]School of GeoSciences, Edinburgh University, Edinburgh, UK

[2]Bristol Glaciology Centre, School of Geographical Sciences, University of Bristol, Bristol,
UK

*Correspondence to Bertie.Miles@ed.ac.uk

**Abstract:** Totten Glacier is currently the largest source of mass loss in the East Antarctic Ice
Sheet. The glacier has been losing mass for decades and inland thinning was detected in the
earliest satellite-altimetry observations in the early 1990s, but for how long the glacier has been
losing mass remains unknown. We calculate decadal ice-speed anomalies to confirm that
Totten Glacier has not undergone sustained acceleration since at least 1973. Together with
observations of grounding-line retreat from 1973-1989, we confirm that the glacier was losing
mass in the 1970s. Surface undulations form on Totten Ice Shelf close to a bathymetric high
point near the grounding line in response to time-varying melt rates and are preserved
downstream for several decades. Using the full Landsat archive, we produce a century-long
record of surface-undulation formation that we interpret as a qualitative record of basal-melt-
rate variability. An anomalous ~20-year absence of undulations associated with the mid-20[th]
century manifests a period when ice passing close to a bathymetric high point near the
grounding line was pervasively thinner, and may represent an anomalous warm period that
triggered the onset of modern-day mass loss at Totten Glacier. We also observe the collapse of
a nearby small ice shelf between 1963 and 1973, that is consistent with a regionalised mid-20[th]
century warm period. Our results highlight that the currently available ~30-year satellite-
altimetry records are not long enough to capture the full scale of decadal variability in basal-
melt rates and mass-loss patterns.

**1. Introduction**

Aurora Subglacial Basin (ASB), located in Wilkes Land, East Antarctica, is drained by five
major outlet glaciers: Holmes, Moscow University, Totten, Vanderford and Denman glaciers
(Fig. 1). Oceanographic observations have confirmed the presence of warm modified
Circumpolar Deep Water (mCDW) on the continental shelf adjacent to each of these outlet
glaciers (Ribeiro et al., 2021; Rintoul et al., 2016; Silvano et al., 2017; van Wijk et al., 2022).
This is accompanied by ongoing grounding-line retreat (Brancato et al., 2020; Li et al., 2023b;
Li et al., 2015; Picton et al., 2023), detachment of ice shelves from pinning points (Miles and
Bingham, 2024) and ice-front retreat (Baumhoer et al., 2021; Miles et al., 2016) identified from
satellite observations. The rate of mass loss of ASB between 2003 and 2019 is estimated at -
20 Gt yr$^{-1}$ (Smith et al., 2020), meaning that mass loss from ASB has accounted for just under
2% of global sea-level rise over this period. Paleo-reconstructions from a variety of geological
evidence show that ASB may have made multi-metre contributions to global sea-level rise

during the warm periods of the mid-Pliocene (Aitken et al., 2016; Cook et al., 2014) when $CO_2$ concentrations were comparable to the present day (Martínez-Botí et al., 2015). Therefore, indications from the past and observations of current mass loss suggest that ASB has the potential to make globally significant contributions to sea-level rise in the coming decades to centuries (Stokes et al., 2022).

Ice discharge from Totten Glacier (Fig. 1) is estimated to be between 65 and 85 Gt yr$^{-1}$ (Davison et al., 2025; Rignot et al., 2019), and is the single largest source of mass loss from both ASB and the entire East Antarctic Ice Sheet (Mohajerani et al., 2018; Nilsson et al., 2022; Rignot et al., 2019; Schröder et al., 2019; Smith et al., 2020). Inland thinning of grounded ice was detected in the early 1990s (Nilsson et al., 2022; Schröder et al., 2019), hinting that the catchment may have been losing mass decades before the earliest satellite-altimetry observations, but the onset date of this current bout of mass loss is unclear. Recent evidence shows that enhanced intrusion of mCDW is the most likely forcing of the mass loss at Totten Glacier. The warm ocean water can flow over the continental shelf (Rintoul et al., 2016) and travel beneath Totten Ice Shelf towards the grounding line via a network of deep bathymetric troughs (Hirano et al., 2023), leading to significant basal melting. However, whether and how this oceanic forcing has varied over time is unknown, leading to large uncertainties in quantifying sea-level rise contributions from Totten Glacier over the coming century and beyond under a variety of different climate-warming scenarios (Jordan et al., 2023; McCormack et al., 2021; Pelle et al., 2021).

Outlet glaciers lose mass through a flux imbalance, whereby ice loss from ice discharge outweighs mass gain from surface mass balance processes. At Totten Glacier, despite its significant mass loss across the satellite era, existing studies show no clear widespread acceleration in ice-flow speed across the Totten system (Li et al., 2023a; Li et al., 2015; Rignot et al., 2019; Roberts et al., 2018). Therefore, the apparent absence of increased ice discharge over the satellite era could be explained by two possibilities: 1) Sparse and lower quality satellite images pre-2000s are not of sufficient quality to capture ice-flow acceleration of the glacier or ice shelf, or 2) There was already a flux imbalance at the onset of the satellite era, meaning the Totten catchment was already losing mass at the onset of satellite observations.

Here, we analyse ARGON and Landsat optical imagery to investigate the behaviour of the Totten glacier and ice-shelf system over the past century. We calculate decadal ice-speed anomalies and grounding-line migration back to 1973. We also produce a century-long record of surface-undulation formation that can be taken as a qualitative record of basal-melt anomalies. We use all three datasets to determine a broad mass-loss history of Totten Glacier.

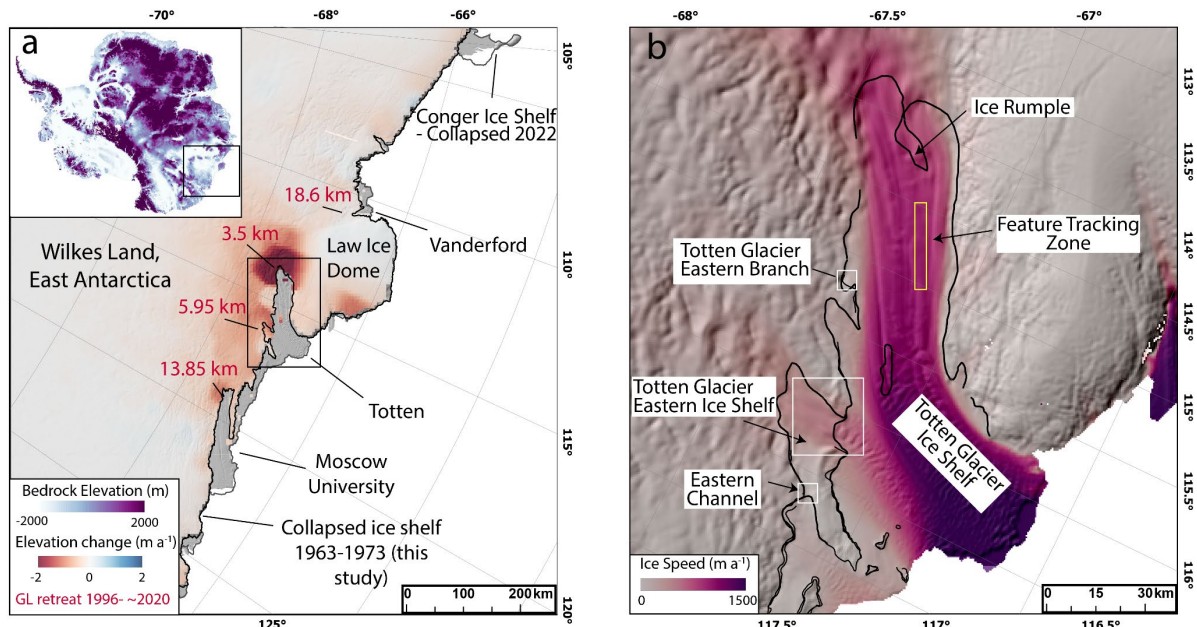

**Figure 1: a)** Ice-surface elevation change in Wilkes Land, East Antarctica between 2003 and 2019 (Smith et al., 2020), with grounding-line retreat (1996~2020) of key outlet glaciers from published studies referenced in the main text annotated in red text. The black line is the MODIS 2009 grounding line (Haran, 2021; Scambos et al., 2007). The inset is bed topography from BedMachine (Morlighem et al., 2020). **b)** Ice speed of Totten system with October 2017 InSAR-derived grounding lines (Floricioiu, 2021) overlain. The yellow box represents the region of manual feature tracking. The white boxes are the regions where grounding line migration is mapped. The background image in both panels is the REMA DEM (Howat et al., 2019).

## 2. Methods

### 2.1 Satellite imagery

The earliest available satellite imagery of Totten glacier system including the ice shelf comprises a pair of ARGON images from 1963 (Fig. 2a). We use the orthorectified and enhanced ARGON imagery of Totten Ice Shelf developed in previous studies (Li et al., 2023a; Ye et al., 2017). The image covering the lower half of the ice shelf and the ice front is of high quality, thus enabling us to identify large crevasses and surface features. However, the image covering the upper half of the ice shelf and grounding line is of relatively poor quality and the identification of surface features is challenging (Fig. 2a). For 1973 and 1989, we use the geo-corrected, cloud-free scenes of Totten Ice Shelf acquired respectively by Landsat-1 and Landsat-4 (Miles and Bingham, 2024). Throughout the 2000s, we were able to exploit more regular imagery from Landsat 7, 8 and 9 (Fig. 2). Sporadic SAR imagery is available throughout the 1990s (e.g. ERS), but was not used in this study because we are mainly interested in surface features on Totten Ice Shelf which can express very differently between SAR and optical imagery.

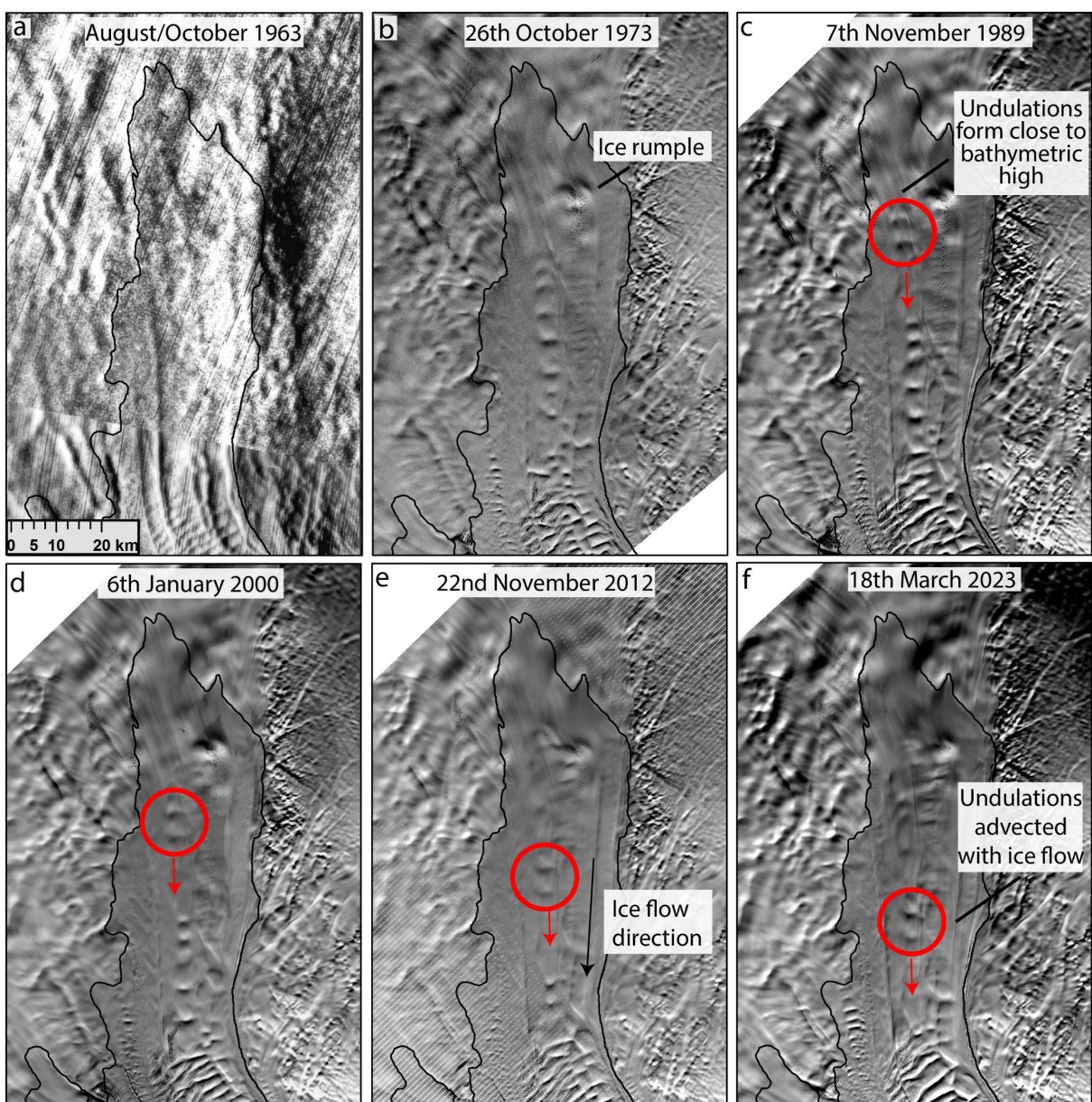

**Figure 2:** Six decades of satellite imagery over Totten Ice Shelf. The red circles highlight surface undulations forming near an ice rumple (location annotated in panel b) in 1989, before being advected downstream with ice flow. ARGON (a) and Landsat (b-f) images are courtesy of the U.S. Geological Survey. Black line shows the MODIS 2009 grounding line (Haran, 2021; Scambos et al., 2007).

## 2.2 Ice Speed

Sporadic and lower quality satellite imagery from before 2000 makes it difficult to determine long-term trends in ice speed at the Totten system. There are two principal reasons for this. Firstly, post-2000 ice-speed records show large interannual variability, with ice speeds varying by ~10% over the course of a few years (Greene et al., 2017; Miles et al., 2022). In the absence of continuous observations in the past, it is difficult to determine whether any apparent changes in ice speed from isolated observations in the past (e.g. 1973, 1989) are related to long-term trends or part of this cyclic interannual variability. Secondly, sparse image availability means that uncertainties can be very high. For example, the only pair of Landsat images covering Totten Glacier between 1975 and 1998 comprises Landsat-4 images from 28[th] March 1989 and 7[th] November 1989. During this relatively short period, ice in the yellow box (Fig. 1b) may

have only flowed ~700 m and, assuming a displacement uncertainty of ±60 m or 2 pixels (1 for co-registration; 1 for feature tracking), this equates to uncertainties of around ±9%. This very high uncertainty could partly explain the vast range (>20%) of ice-speed anomaly estimates for Totten Glacier in 1989 reported in previous studies (Li et al., 2023a; Li et al., 2015; Rignot et al., 2019; Roberts et al., 2018).

To address both of these issues, we manually track surface features on Totten Ice Shelf using image pairs separated by much longer periods of time (>10 years), which both smooths the impact of interannual variability and reduces relative feature tracking uncertainties because surface features are travelling much greater distances. For example, between 1973 and 1989 we track a 16.07 km displacement of a surface feature and, assuming a displacement uncertainty of one pixel for co-registration and two pixels for feature tracking or 180 m in total (60 m pixel size for Landsat-1), this equates to a smaller uncertainty of ±1.14% (Fig. 3). The disadvantage of this method is that very few surface features remain visible over this length of temporal separation and the process of tracking features over long periods of time can systematically overestimate ice-flow speed if ice does not flow in a straight line (Li et al., 2022). After a careful inspection of the imagery, we focus our feature tracking on one prominent feature, of unknown origin, that repeatedly forms around 20 km downstream (Location; yellow box, Fig. 1b) of the grounding line, and is the only feature that unambiguously maintains its shape and structure for multiple decades (Fig. 3). We report changes in ice speed in each epoch as an anomaly with respect to the MEaSUREs ice-velocity reference mosaic of Antarctica (Rignot et al., 2011). This is because, in each epoch, the location of the feature that is tracked starts in a slightly different position. All relative anomalies are calculated along the same flowline and the maximum distance between the centre point of any two anomalies is <20 km. Therefore, given their close proximity it is reasonable to infer that the anomalies are closely comparable to each other through time. In total, we measure decadal ice-speed anomalies from 10 epochs detailed in Table 1, we do not prescribe a 1-pixel co-registration uncertainty for image pairs containing the higher quality Landsat-7 data onwards

We also calculate interannual ice-speed anomalies at Totten Ice Shelf from 1989 to 2023 where there is a greater availability of satellite imagery (Table 2). We calculate interannual ice-speed anomalies by manually tracking the same surface feature used in decadal ice-speed anomalies. For these image pairs, we prescribe a feature tracking uncertainty of 1 pixel because the images are more closely separated in time and an additional 1-pixel co-registration uncertainty for all pre-Landsat-7 images. To investigate whether ice-speed anomalies within the feature-tracking zone serve as reasonable proxies for ice-speed anomalies at the grounding line, we use the MEaSUREs annual velocity mosaics (Mouginot et al., 2017). We compare ice-speed anomalies within the feature-tracking zone to ice-speed anomalies within a ~15×5 km bounding box at the grounding line (Fig. 5) in all years for which data are available in both locations (2008–2009, 2009–2010, and 2013–2020).

**Table 1:** Feature tracking: decadal ice-speed anomalies. Uncertainty is assumed to be three pixels—one from image co-registration and two from feature tracking.

| Image 1 | Image 2 | Days | Distance (km) | Average speed (m a⁻¹) | MEaSUREs speed (m a⁻¹) | Anomaly (%) | Uncertainty (km) | Uncertainty (%) |
|---|---|---|---|---|---|---|---|---|
| 26/10/1973 | 07/11/1989 | 5856 | 16.07 | 1002 | 976 | 2.7 | 0.18 | 1.14 |
| 07/11/1989 | 06/01/2000 | 3712 | 9.54 | 938 | 947 | -1.0 | 0.09 | 0.94 |
| 06/01/2000 | 03/12/2010 | 3984 | 10.70 | 980 | 974 | 0.6 | 0.06 | 0.56 |
| 22/12/2002 | 22/11/2012 | 3623 | 10.01 | 1008 | 981 | 2.8 | 0.06 | 0.62 |
| 22/11/2006 | 25/11/2016 | 3656 | 10.07 | 1005 | 997 | 0.8 | 0.06 | 0.60 |
| 09/11/2007 | 25/09/2017 | 3608 | 9.89 | 1003 | 999 | 0.4 | 0.06 | 0.60 |
| 29/12/2008 | 17/12/2018 | 3640 | 9.98 | 1001 | 1005 | -0.4 | 0.06 | 0.60 |
| 14/11/2009 | 30/08/2019 | 3576 | 9.83 | 1003 | 1009 | -0.6 | 0.06 | 0.61 |
| 03/12/2010 | 19/10/2020 | 3608 | 9.88 | 999 | 1015 | -1.6 | 0.06 | 0.60 |
| 17/11/2013 | 24/03/2023 | 3414 | 8.86 | 947 | 959 | -1.3 | 0.06 | 0.67 |

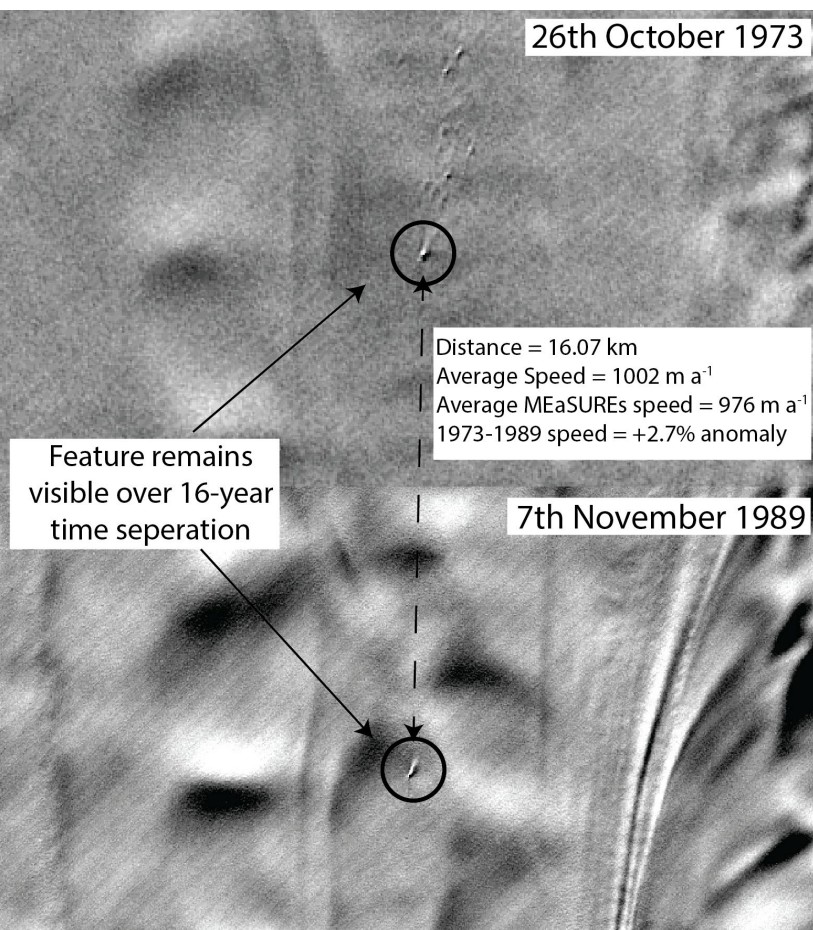

**Figure 3:** Image split showing an example of manual feature tracking from a Landsat-1 image from 26th October 1973 to a Landsat-4 image from 7th November 1989. See Figure S1 for more examples. Landsat images are courtesy of the U.S. Geological Survey.

**Table 2.** Feature tracking: interannual ice-speed anomalies. Uncertainty is assumed to be two pixels for 1989—one from image co-registration and one from feature tracking—and one pixel for 2000 onwards, associated with feature tracking.

| Image 1 | Image 2 | Days | Distance (km) | Average speed (m a$^{-1}$) | MEaSUREs speed (m a$^{-1}$) | Anomaly (%) | Uncertainty (km) | Uncertainty (%) |
|---|---|---|---|---|---|---|---|---|
| 28/03/1989 | 07/11/1989 | 224 | 0.69 | 1124 | 1004 | 12.0 | 0.06 | 8.70 |
| 06/01/2000 | 22/12/2002 | 1081 | 2.76 | 931 | 963 | -3.3 | 0.03 | 1.09 |
| 22/12/2002 | 22/11/2006 | 1431 | 3.85 | 981 | 971 | 1.0 | 0.03 | 0.78 |
| 22/11/2006 | 29/12/2008 | 768 | 2.14 | 1017 | 980 | 3.8 | 0.03 | 1.40 |
| 29/12/2008 | 03/12/2010 | 704 | 1.98 | 1029 | 985 | 4.5 | 0.03 | 1.51 |
| 03/12/2010 | 22/11/2012 | 720 | 2.00 | 1014 | 1000 | 1.4 | 0.03 | 1.50 |
| 22/11/2012 | 19/10/2014 | 696 | 1.85 | 972 | 1005 | -3.3 | 0.03 | 1.62 |
| 19/10/2014 | 25/11/2016 | 768 | 2.10 | 998 | 1016 | -1.8 | 0.03 | 1.43 |
| 25/11/2016 | 17/12/2018 | 752 | 2.06 | 1001 | 1024 | -2.2 | 0.03 | 1.45 |
| 17/12/2018 | 19/10/2020 | 672 | 1.89 | 1026 | 1034 | -0.8 | 0.03 | 1.59 |
| 19/10/2020 | 18/03/2023 | 880 | 2.52 | 1046 | 1044 | 0.2 | 0.03 | 1.19 |

## 2.3 Grounding Line

We use Landsat imagery to map the break-in-slope (Christie et al., 2016; Fricker et al., 2009), a proxy for the grounding-line location. Our analysis is limited to three locations where we observe clear breaks-in-slope, which are the Totten Glacier Eastern Branch, Totten Glacier Eastern Ice Shelf and the Eastern Channel (Fig. 1b). In these regions, breaks-in-slope are a good representative for the true grounding line and have been previously used in analysing the basal-melting regimes in Totten Ice Shelf due to warm ocean water intrusions (Greenbaum et al., 2015; Li et al., 2023b). We do not map the grounding line across Totten Glacier's main trunk over which ice flow is fast and consequently the break-in-slope is an unreliable proxy for the actual grounding line. In this location, grounding-line detection is also challenging to measure using DInSAR interferograms or satellite-altimetry datasets (Li et al., 2015; Li et al., 2023b). We map the break-in-slope in each location in 1973, 1989, 2000 and 2023. Based on previous studies, we prescribe a mapping uncertainty of 1 pixel (Christie et al., 2016), along with an additional 1 pixel uncertainty associated with image co-registration.

## 2.4 Surface Undulations

Surface undulations downstream of bedrock obstacles on floating ice shelves are common features observed across several ice shelves, with prominent examples expressed on Filchner-Ronne (Brunt et al., 2011), Pine Island (Bindschadler et al., 2011), Thwaites (Kim et al., 2018), Totten (Roberts et al., 2018), and Ross (Lee et al., 2012) ice shelves. As ice flows over bedrock obstacles, the topographic variability of the bedrock surface is transmitted to the glacier surface (De Rydt et al., 2013; Gudmundsson, 2003). If the thickness of floating ice varies over time in response to time-varying basal-melt rates, the degree of contact with the underlying bedrock obstacle will also change, resulting in variable imprints on the surface topography of the ice shelf that are subsequently transported downstream with ice flow.

Sporadic trains of surface undulations form on Totten Ice Shelf close to a large ice rumple near the grounding line (Fig. 2; Animation S1). These features have been hypothesised to develop

in response to time-varying basal-melt rates, whereby lower melt rates lead to enhanced grounding on a bathymetric high, a reduction in ice speed, and the formation of undulations (Roberts et al., 2018). An inspection of our cloud-free Landsat imagery spanning 50 years reveals that the trains of undulations persist relatively unchanged in shape for ~45 km downstream of the ice rumple as they are advected with ice flow. Therefore, in our earliest cloud-free image from 1973, visible surface undulations at the furthest point downstream of the ice rumple formed decades earlier (Fig. 4) and our full time series provides a near century-long insight into surface-undulation formation.

To further investigate how the surface undulations form on Totten Ice Shelf, we first measure surface-elevation anomalies from ICESat-2 repeat tracks across the ice shelf to capture the dimensions of the undulations as they are advected downstream with ice velocity (Li et al., 2023b). We use the ICESat-2 ATL06 dataset for Track 26 between May 2019 and November 2023. The ICESat-2 data are applied using the 'ATL06_quality_summary' flag to remove poor-quality elevation measurements possibly caused by clouds or high surface roughness (Smith et al., 2019). They are further corrected for the ocean tide and ocean-loading tide, as well as inverted barometer effects (DAC). Instead of using the ocean-tide correction values generated from the GOT4.8 model that are provided in the ATL06 data product, we use the CATS2008 tidal model to estimate the ocean-tide corrections (Padman et al., 2002).  Elevation anomalies are calculated by subtracting a reference elevation profile, defined as the average elevation from each individual repeat track elevation profile. We also calculate anomalies in the rate of grounded-ice thinning over a 5-year rolling window within a 40 x 40 km box upstream of the grounding line from 1985-2019, using the Antarctic Ice Sheet Elevation Change data (Nilsson et al., 2022) provided by the NASA MEaSUREs ITS_LIVE project (Nilsson et al., 2025). To calculate surface mass balance anomalies between 1979 and 2024, we use the Regional Atmospheric Climate Model (RACMO), version 2.4p1 (Van Dalum et al., 2025). For basal-melt variability, we use the in situ basal-melt time series (2017–2019) derived from autonomous phase-sensitive radar (TI04 – Vaňková et al., 2023), located 5 km upstream of the region where surface undulations form. To examine where surface undulations form in relation to bathymetric highs, we use two bathymetry products. First, we use BedMachine (Morlighem et al., 2020), which incorporates the Greenbaum et al. (2015) bathymetry model as its basis at Totten. Second, we use the updated bathymetry model of Vaňková et al. (2023), which includes additional gravimetry and seismic surveys near the grounding line. We examine both datasets because of differences in bathymetry near the region where surface undulations develop.

We then create a time series of surface-undulation formation by estimating the date on which each undulation formed at the ice rumple. This is estimated by calculating the average flow speed from the MEaSUREs ice-velocity mosaic of Antarctica along a flowline from the ice rumple to 45 km downstream, which represents the region where surface undulations are visible before they are lost to a crevasse field. However, because melt rates are highest near the grounding line (Adusumilli et al., 2020), it is the cumulative melt as ice travels downstream from the grounding line that determines the thickness of ice as it flows over the ice rumple (Fig. 4). Therefore, we also estimate the date on which ice flowed over the grounding line by calculating the average flow speed from the MEaSUREs ice-velocity mosaic of Antarctica along a flowline from the grounding line to the ice rumple.

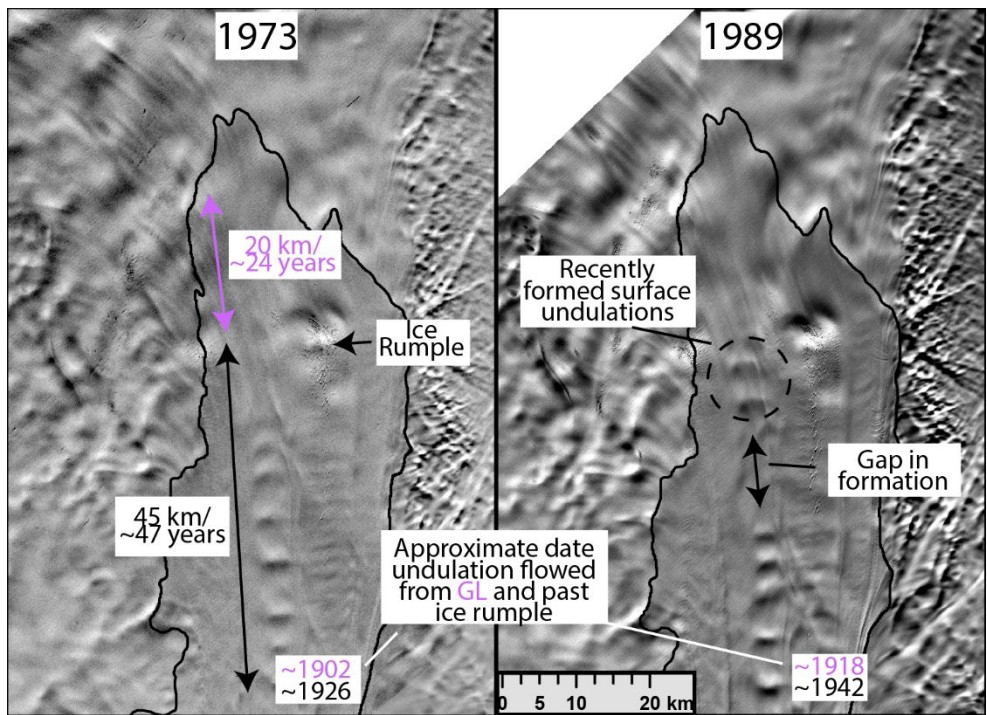

**Figure 4:** Landsat imagery from 1973 and 1989 showing the time taken for ice to flow from the grounding line to the ice rumple and from the ice rumple to the most distant surface undulation. Landsat images are courtesy of the U.S. Geological Survey

## 3. Results

### 3.1 Ice Speed

Ice-speed anomalies within the feature-tracking zone on Totten Ice Shelf are correlated (p-value = 0.02) to ice-speed anomalies at the grounding line (Fig. 5a and b). Therefore, our multidecadal measurements of ice speed anomalies derived from the feature-tracking zone 20 km downstream of the grounding line are reasonable proxies for ice speed anomalies at the grounding line. We observe no clear trend in multidecadal ice-speed anomalies at Totten Ice Shelf between 1973 and 2023, with ice-speed anomalies ranging from -1.6 ±0.6% from 2010-2020 to +2.8 ±0.62% from 2002-2012 (Fig. 5). We observe a much greater range of ice-speed anomalies over interannual timescales. We record a maximum ice speed anomaly of +12 ±8.7% in 1989 and a minimum ice speed anomaly of -3.3 ±1.1% from 2000-2002 (Fig. 5). Regular ice-speed measurements between 2000 and 2023 reveal oscillatory behaviour in speed anomalies of up to ~8% over the course of 6-7 year cycles (Fig. 5). There is no evidence of a sustained speed-up of Totten Glacier over the past 50 years.

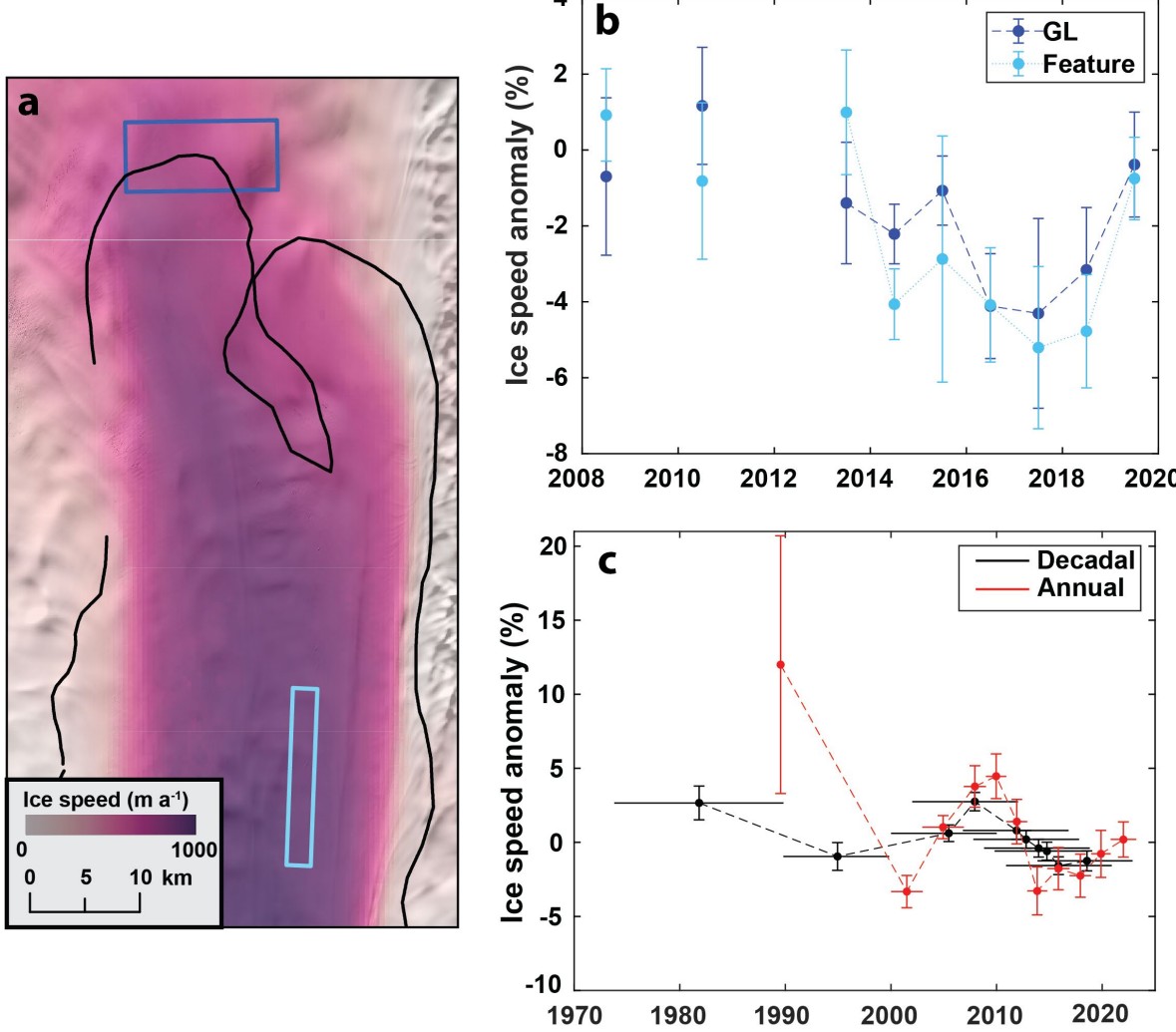

**Figure 5:** Ice-speed anomalies at Totten Ice Shelf. **a)** Location of bounding boxes used to extract annual ice-speed anomalies at the grounding line (blue) and feature-tracking zone (light blue), with ice speed from the MEaSUREs ice velocity mosaic and grounding lines from October 2017 overlain (Floricioiu, 2021). **b)** Annual ice-speed anomalies from the MEaSUREs annual mosaics. **c)** Decadal and annual ice-speed anomalies extracted manually from within the feature box. Black data points represent decadal ice-speed anomalies; red data points represent the interannual ice-speed anomalies. Horizontal lines represent the time gap between the two images used to track surface features.

## 3.2 Grounding Line

At the Totten Glacier Eastern Branch, we observe a maximum $4.1 \pm 0.12$ km grounding-line retreat between 1973 and 2023, that includes a $0.8 \pm 0.12$ km maximum retreat between 1973 and 1989 (Fig. 6a). The width of the mouth of the Totten Glacier Eastern Ice Shelf increased by around $3.8 \pm 0.12$ km between 1973 and 2023 with retreat on both the northern and southern flanks (Fig. 6b). The rate of retreat is reasonably consistent in each of our three epochs: 1973-1989; 1989-2000 and 2000-2023. At Eastern Channel we observe a gradual widening through each of our 3 epochs, and between 1973 and 2023 the channel width approximately doubled (Fig. 6c).

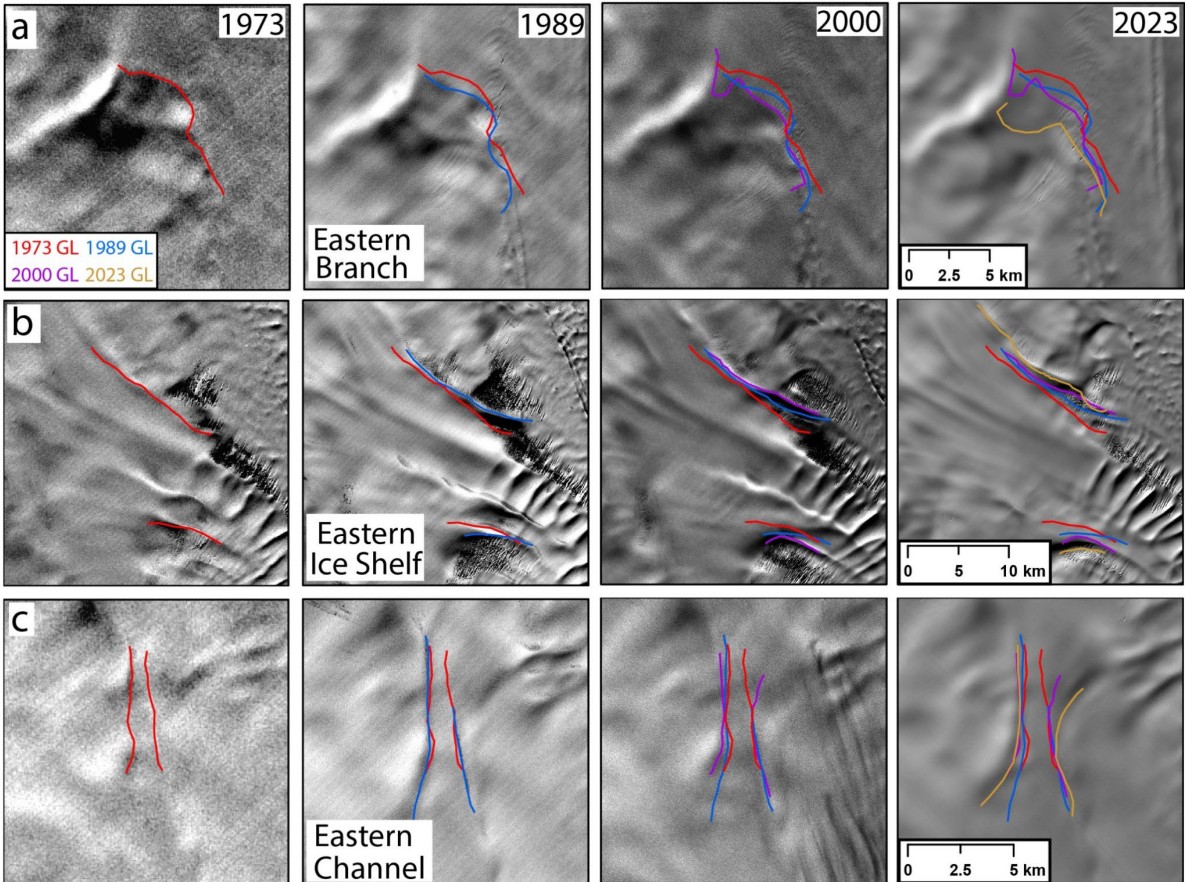

**Figure 6:** Grounding-line retreat tracked from Landsat imagery for **a)** Eastern Totten Ice Shelf, **b)** Eastern Ice Shelf and **c)** Eastern Channel (location of each panel is shown as white boxes in Fig. 1b). Landsat images are courtesy of the U.S. Geological Survey.

## 3.3 Surface Undulations

### 3.3.1 Surface-undulation formation

ICESat-2 repeat tracks across a prominent surface undulation as it flows downstream show a surface elevation change of up to 25 m over three years, or approximately 8 m a⁻¹ (Fig. 7d). The magnitude of this elevation anomaly is broadly comparable to in situ estimates of basal-melt variability (~9 m a⁻¹), measured 5 km upstream of the location where the surface undulations form (Fig. 7e). Between 1985 and 2019, grounded ice within a $40 \times 40$ km bounding box immediately upstream of the grounding line thinned at an average rate of approximately 1 m a⁻¹, with interannual variability typically within ±0.5 m a⁻¹ (Fig. S2). Mean surface mass balance between 1979 and 2024 is approximately 1 m a⁻¹, with interannual variability in the range of ±0.4 m a⁻¹ (Fig. S3).

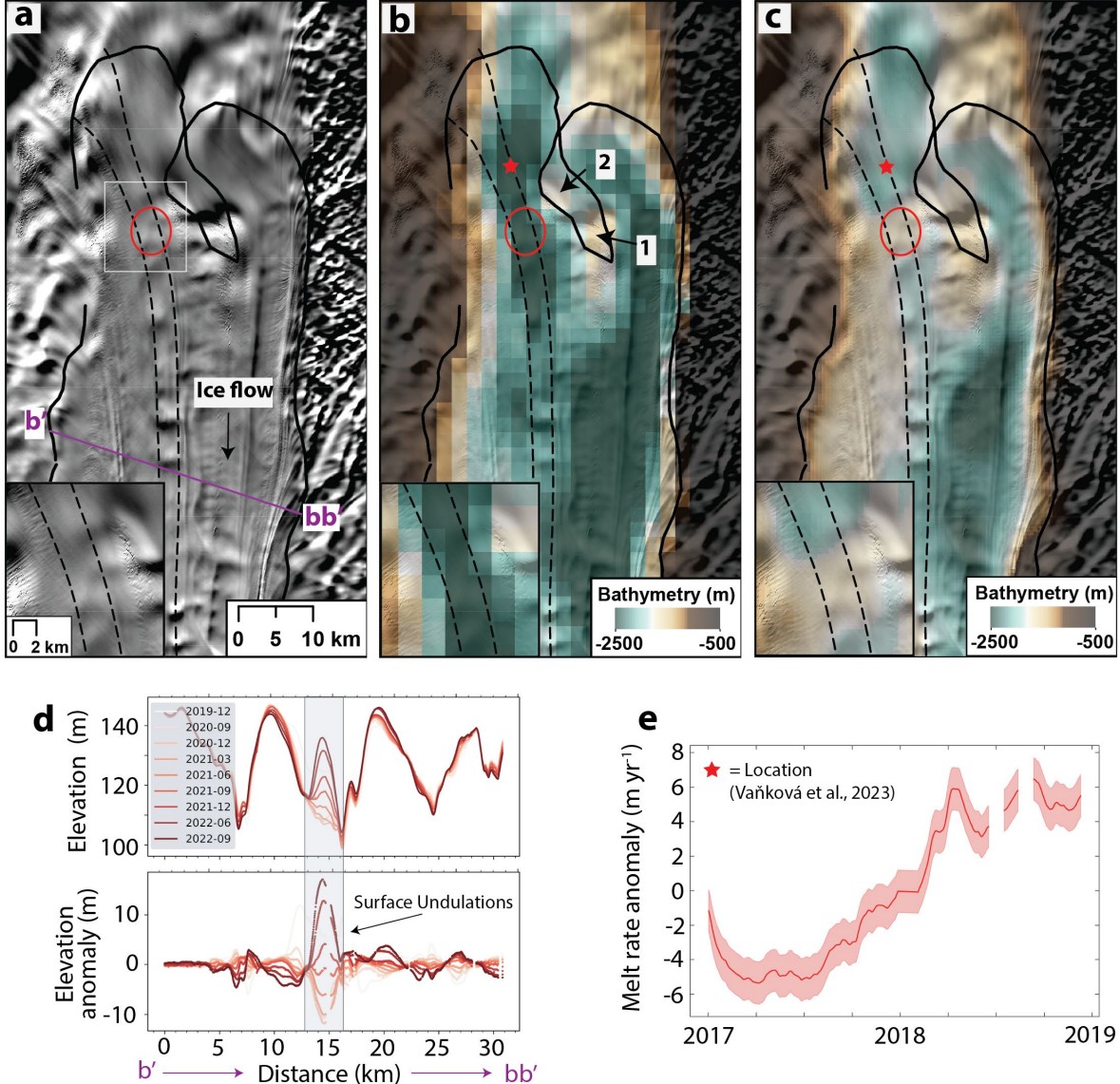

**Figure 7: a)** Landsat-8 image from 17th November 2013 showing recently forming surface undulations. The white box is the inset for the zoomed inset, bottom left. The purple transect represents the profile of the ICESat-2 repeat tracks. **b)** Vaňková et al. (2023) bathymetry. **c)** BedMachine bathymetry (Morlighem et al., 2020). The dotted lines on panels a-c represent flow lines of the surface undulations, the red circle represents the location where surface undulations are first visible in optical imagery, and the black line is the grounding lines derived from October 2017 (Floricioiu, 2021). Label (1) points to the main ice rumple, label (2) points to the secondary ice rumple. **d)** Repeat ICESat-2 tracks over Totten Ice Shelf highlighting the surface-elevation anomalies associated with the surface undulations as they flow downstream. **e)** In situ basal-melt timeseries from Vaňková et al. (2023) located at the red star.

### 3.3.2 Surface-undulation timeseries

Based on the average flow speed of 950 m a⁻¹ along the transect of the surface undulations, and assuming ice-flow speed has remained constant through time, the most distant surface undulation 45 km downstream of the ice rumple will have formed ~47 years before the image date (Fig. 4). We also calculate that it takes approximately 24 years for ice to flow from the grounding line to the ice rumple based on an average flow speed of 850 m a⁻¹ (Fig. 4). This means that in our earliest Landsat image in 1973, the most distant surface undulation will have

formed in approximately 1926, being imprinted into ice that had crossed the grounding line in approximately 1902 (Fig. 4). The time series of surface-undulation formation reveals three distinct phases of surface-undulation formation over the past 100 years (Fig. 8). For simplicity, we define the start and end point of each phase by the dates on which ice crossed the ice rumple, but note that these dates would be shifted ~20 years earlier in time if the phases were defined by the dates on which the same ice crossed the grounding line. The first phase is characterised by regular surface-undulation formation at the ice rumple between 1920 and 1965. The second phase is characterised by a gap in surface-undulation formation between 1965 and 1980. The third phase is characterised by intermittent formation of surface undulations between 1980 and 2020. Notably, surface undulations in this final phase appear to reduce progressively in size, possibly implying a progressive thinning of the ice shelf.

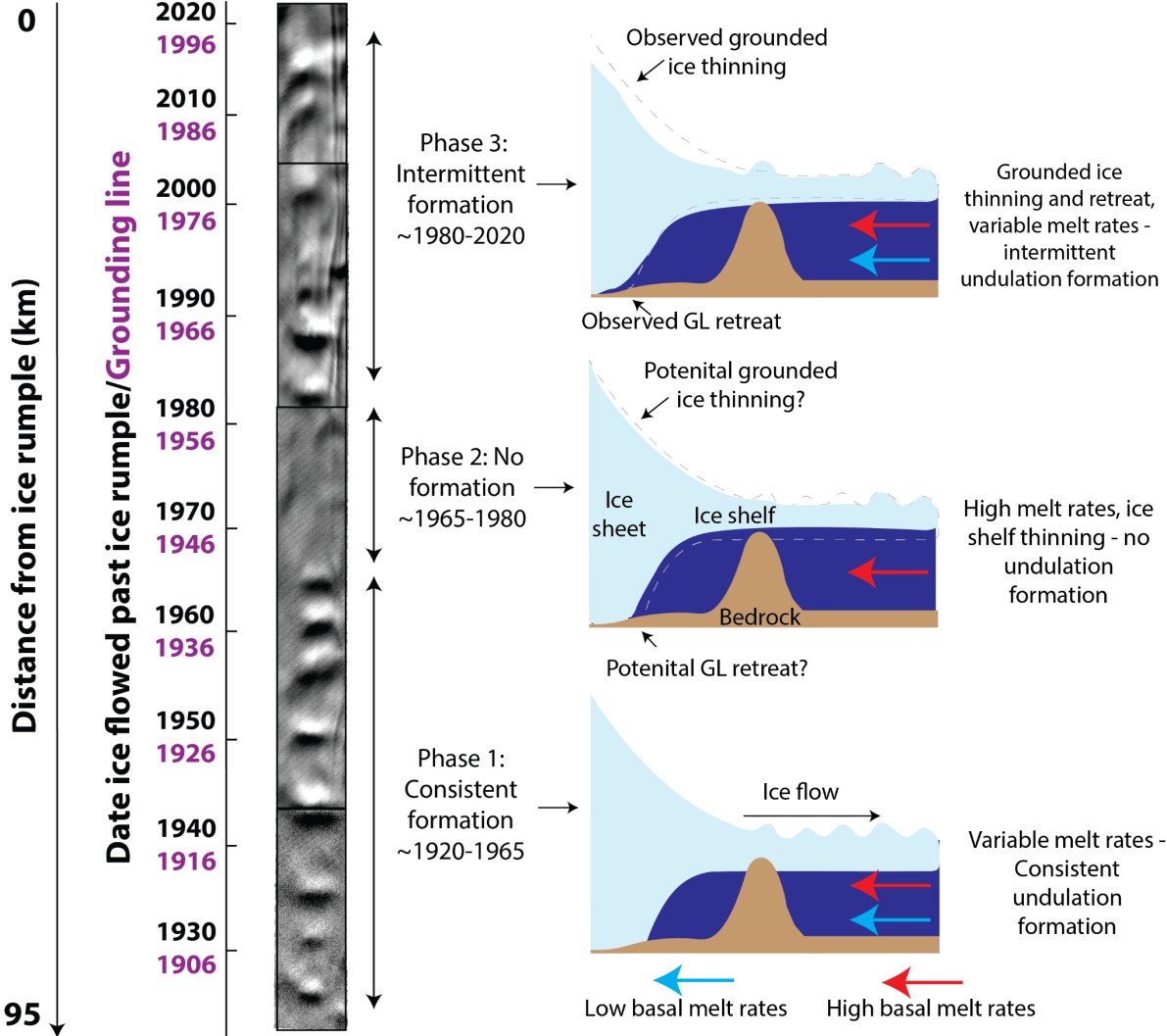

**Figure 8:** Time series of surface-undulation formation comprised of four Landsat images (1973, 1989, 2000 and 2023) stitched together (image boundaries denoted by black boxes). The time series is separated into three phases, with a corresponding schematic showing the processes driving surface-undulation formation.

## 4. Discussion

### 4.1 No speed-up of Totten Glacier since at least 1973

Our results show that there has been no clear multidecadal trend in ice speed at Totten Ice Shelf from 1973-2023 (Fig. 5), implying that there have also been no sustained changes to ice-shelf buttressing. There are no indications of any major changes in the thickness of ice at the grounding line over the past 50 years. The grounding line of the main Totten outlet has retreated up to a maximum of 3 km along a relatively flat slope of -0.3° between 1996 and 2013 (Li et al. 2015), accounting for an approximately 15 m deepening in bed elevation. Grounded ice is thinning at a rate around 0.5 m a$^{-1}$ (Fig. S2), equivalent to 25 m of thinning near the grounding line over the past 50 years. Ice flowing over the grounding line is about 2300 m thick (Li et al., 2015), thus any changes in ice thickness flowing over the grounding line have been proportionally negligible given Totten's current ~10% imbalance (Rignot et al., 2019). This, combined with limited changes in ice speed, means that average ice discharge has likely remained relatively unchanged at Totten Glacier over the past 50 years, despite strong interannual variability. Although there exists some interdecadal variability in snowfall (Kim et al., 2024), there is no evidence for any long-term trends in snowfall from the nearby Law Dome ice core (Jong et al., 2022; Roberts et al., 2015) and, because Totten's present ice discharge outweighs input from snowfall, this means that the Totten catchment was likely losing a similar amount of mass from 1973-1989 in comparison to the 2000s. Thus, the trigger for mass loss at Totten catchment predates the onset of the Landsat satellite record.

We note that Li et al. (2023a) inferred a long-term speed up at the Totten grounding line between 1963-1973 and 2022, but with very little change in ice speed 50 km upstream of the ice front over the same time period (Fig. 2; Li et al., 2023a). However, the rate of acceleration in ice speed at the grounding line reported in Li et al. (2023a) was not consistent over this time period, with a large ~18% acceleration between 1963-1973 and 1973-1989, then a limited change in ice speed between 1973-1989 and 2022. From our methods, we can only reiterate that the Totten system has not experienced any multidecadal acceleration since 1973, because we were unable to track any surface features on the 1963 ARGON image near the grounding line.

### 4.2 Ongoing grounding-line retreat since at least 1973

Our results show consistent grounding-line retreat at Totten Glacier Eastern Ice Shelf and Totten Glacier Eastern Branch between 1973-1989, 1989-2000 and 2000-2023 (Fig. 6). Our observation that the grounding line was already retreating in regions where the break-in-slope is visible between 1973 and 1989 further supports the notion that the Totten Glacier system was already out of balance and losing mass during this period. The Eastern Channel has also been consistently widening from 1973-1989, 1989-2000 and 2000-2023 (Fig. 6c). This channel, first discovered by radio-echo sounding (Greenbaum et al., 2015), has subsequently been shown to be tidally modulated via satellite-radar interferometry (Li et al., 2023b). The Eastern Channel is important because it may provide a direct pathway for warm mCDW to enter the Eastern Ice Shelf cavity (Greenbaum et al., 2015). The expanding width of the channel since at least 1973 would suggest a greater volume of tidally pumped warm mCDW is being progressively funnelled through this trough into the ice-shelf cavity.

### 4.3 Surface-undulation formation

Surface undulations on Totten Ice shelf form in response to an intermittent local re-grounding of ice driven by time-varying basal-melt rates. This is because: 1) They form on or downstream of bathymetric highs. 2) Their surface signature is incompatible with other features (e.g., basal crevasses; Luckman et al., 2012; McGrath et al., 2012) that can form downstream of pinning points, and 3) The elevation anomaly associated with the surface undulations broadly matches observations of basal-melt variability (Fig. 7).

The bathymetry near the Totten grounding zone is complex and despite recent improvements, uncertainties remain (Fig. 7b versus c). On the basis of updated seismic and gravimetry-derived bathymetry of the Totten grounding zone (Vaňková et al.,2023), surface undulations form in a narrow (3-4 km wide) channel between ice grounded on the eastern flank of the ice shelf and a secondary bathymetric high point upstream of the main ice rumple (Fig. 7b, Animation S1). In this scenario, surface undulations could form during periods of greater localised re-grounding on the secondary bathymetric high causing a narrowing of the channel. However, an earlier bathymetry product (Greenbaum et al., 2015) resolved a much more prominent broad bathymetric ridge directly under the zone of surface-undulation formation (Fig. 7c). Although the presence of this broad ridge was deemed highly incompatible with the Vaňková solution, Vaňková et al. (2023) acknowledged that uncertainties remain with their more recent product. Therefore, an alternative explanation is that surface undulations form in response to a localised re-grounding on a bathymetric high, smaller in prominence than inferred by the Greenbaum solution, directly underneath where the surface undulations first become visible in optical imagery (Fig. 7).

The surface undulations that we have tracked along Totten Ice Shelf are large features that result in up to 25 m of differential surface-elevation anomalies over the course of three years (Fig. 7d). In situ observations of basal-melt rates from autonomous phase-sensitive radars ~5 km upstream of the undulation-formation zone have shown basal-melt variability of around 9 m a$^{-1}$ (Vaňková et al., 2023; Fig. 7e), the magnitude of which compares well to the surface-elevation anomalies of the undulations. Furthermore, because the in situ basal-melt timeseries is short (two years), the full range of basal-melt variability may not have been captured. In comparison, the interannual variability associated with grounded-ice elevation change and surface mass balance (Fig. S2) is an order of magnitude lower. Negative strain rates have also been observed in the surface undulations formation zone (Roberts et al., 2017). At least in the 2000s when regular satellite imagery was available, undulations formed in circa 2001 and 2012-2015 (Animation S1), coinciding with negative ice speed anomalies (Figure 5c). Therefore, while basal melt variability is controlling the temporal formation of the surface undulations, ice dynamics is also playing an important role in their formation as ice locally slows down in response to increased basal drag.

The physical mechanism proposed to drive the large interannual variability in basal-melt rates is variability in the depth of the thermocline on the continental shelf (Hirano et al., 2023). Oceanographic profiles have confirmed a two-layered structure of the water column with cold Winter Water overlying warm mCDW, separated by a sharp boundary (Hirano et al., 2023; Nakayama et al., 2023; Rintoul et al., 2016). Updated bathymetry on the continental shelf has shown that the depth of the thermocline between the two water masses broadly matches the depth of the shallowest sill on the Central Totten Trough that funnels warm mCDW towards the ice shelf (Hirano et al., 2023). Therefore, small changes in the depth of the thermocline could cause large differences in the temperature of water transported beneath Totten Ice Shelf,

driving large variability in basal-melt rates and surface-undulation formation. Repeat oceanographic observations and moorings show variability in thermocline depth on both interannual and tidal timescales, but these observations are limited to just four years (Rintoul et al., 2016; Hirano et al., 2023). However, observations from satellite altimetry since the early 1990s show multi-metre variations in ice-shelf thickness anomalies, the magnitude of which can only be explained by basal-melt variability (Roberts et al., 2018), suggesting that variability in basal-melt rates has persisted for decades. Roberts et al. (2018) calculated that the wavelength (6-7 years) of ice-shelf thickness variations matches the wavelength of the surface undulations in a Landsat-7 image from 2010, further supporting the notion that the surface undulations are formed by time-varying melt rates. A number of different processes could contribute to driving variability in the thermocline depth, including variations in Antarctic Slope Current strength (Nakayama et al., 2021), intrinsic variability (Gwyther et al., 2018), zonal wind strength on the continental-shelf break (Greene et al., 2017) and coastal-polynya activity (Gwyther et al., 2014; Khazendar et al., 2013), but the dominant driver remains unclear.

**4.4 Anomalously high basal-melt rates in the mid-20th century**

The surface-undulation timeseries provides a novel proxy for basal-melt rates and ice-shelf thickness variability over the past century. In Phase 1, dating from ~1920-1965, surface-undulation formation is consistent (Fig. 8), implying cyclic warm and cool anomalies in basal-melt rates driving intermittent re-grounding on bathymetric highs. In Phase 2, dating from ~1965-1980, no surface undulations form (Fig. 8). This can be explained by a prolonged period of increased basal-melt rates and consequently thinner ice passing over the bathymetric high, possibly driven by a shallowing of the thermocline on the continental shelf. In Phase 3, dating from ~1980-2020, surface undulations form intermittently (Fig. 8). A key implication here is that, on average, melt rates must have been lower during ~1980-2020 (Phase 3) relative to 1965-1980 (Phase 2). This is because surface undulations still form in Phase 3 despite ongoing grounded-ice thinning (Fig. S2) that causes progressively thinner ice to flow into the ice shelf, meaning that progressively more extreme cool anomalies are required for surface-undulation formation (Fig. 8). There is also the possibility of progressively warmer CDW being fed onto the continental shelf since the early 20th century in response to a wind-driven southward shift of the Antarctic Coastal Current (Herraiz-Borreguero and Naveira Garabato, 2022; Yamazaki et al., 2021). In addition, the ongoing grounded-ice thinning could explain why surface undulations also appear to be visibly smaller in ~1980-2020 (Phase 3) compared to ~1920-1965 (Phase 1).

Modern satellite-altimetry observations of ice-shelf thickness change at Totten Ice Shelf support our hypothesis that the absence of surface-undulation formation in Phase 2 represents a period of higher basal-melt rates and relatively thinner ice passing over bathymetric high points near the ice rumple. Observations of ice-shelf thickness change between 2003 and 2019 (Smith et al., 2020) show a conflicting thickness-change signal across the ice shelf (Fig. 9b). The upper ice shelf near the ice rumple thickened, while the mid-section of the ice shelf thinned and there was limited change in the lower ice shelf during this period. This unusual pattern can be explained by the Eulerian advection of relatively thinner and thicker bands of ice downstream with ice flow. Basal-melt rates are an order of magnitude higher near the grounding line (>50 m yr$^{-1}$), compared to the mid-ice shelf (~10 m yr$^{-1}$) (Fig. 9a; Adusumilli et al., 2020). This means that variability in melt rates would be expected to be larger near the grounding line compared to the mid-ice shelf. Therefore during ~1965-1980 (Phase 2) when

basal-melt rates were higher and no surface undulations formed, ice flowed out of the zone of high basal-melt rates (i.e. grounding zone) and was relatively thinner, compared to Phase 1 where basal-melt rates were lower and undulations formed consistently. Thus, as the band of ice associated with high basal-melt rates at the grounding line and ice rumple in 1965~1980 flows downstream, it will replace relatively thicker ice from Phase 1 and cause a thinning signal. A similar process can explain the thickening in the upper ice shelf. Relatively cooler melt rates in Phase 3 (~1980-2020) result in thicker ice flowing out of the zone of higher basal-melt rates, replacing the thinner ice associated with Phase 2, causing a thickening signal. This hypothesis is supported by the fact that the length of the core thinning section (~20 km) in the ice-thickness-change dataset (Fig. 9b) broadly matches the displacement of ice expected over the 16-year time-separation (~16 km in 2003-2019), as well as the length of Phase 2 during which no surface undulations formed (~18 km; Fig. 8). Secondly, the downstream boundary of the thinning section of the ice shelf corresponds directly to the boundary between Phase 1 and Phase 2 of undulation formation, while the upstream boundary of the thinning section corresponds directly to the boundary between Phase 2 and Phase 3 of undulation formation (Fig. 9b).

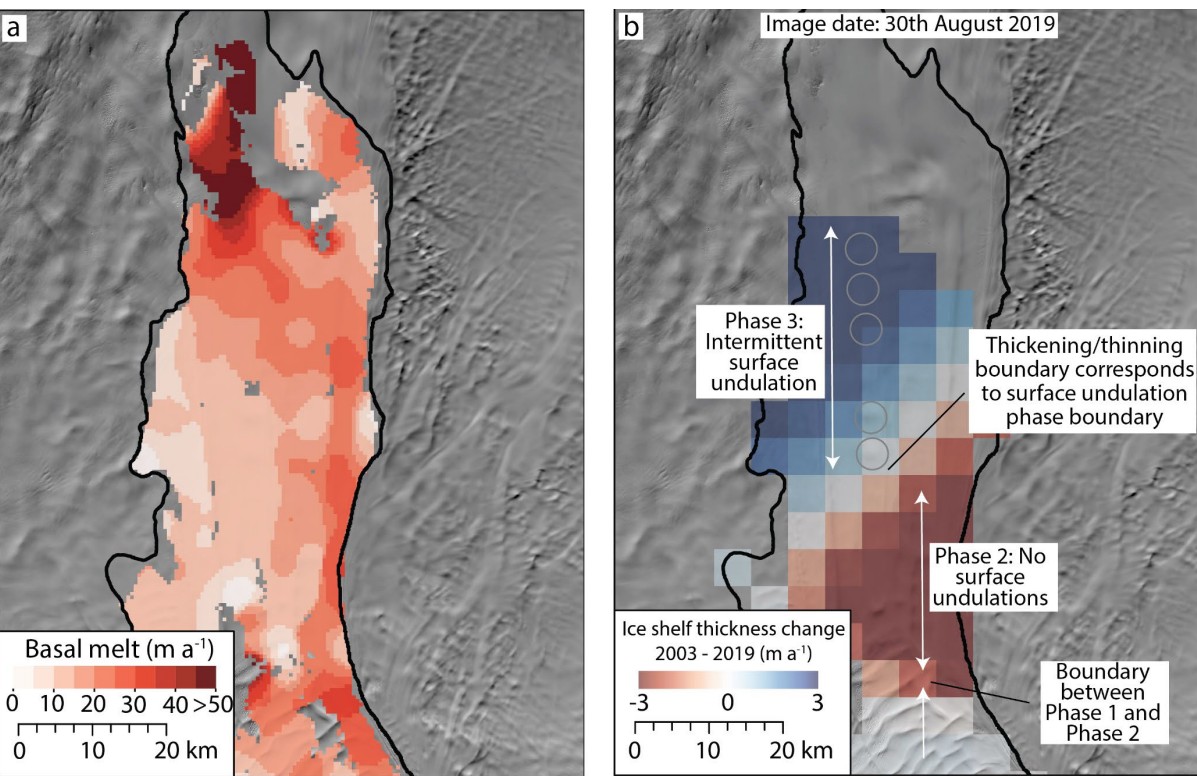

**Figure 9: a)** Basal-melt rates averaged from 2010-2018 **(**Adusumilli et al., 2020). b) Ice-shelf thickness change between 2003 and 2019 (Smith et al., 2020) overlain on a Landsat-8 image from August 2019. The grey circles highlight visible surface undulations in the Landsat image from August 2019. Black line on both images is the MODIS 2009 grounding line (Haran, 2021; Scambos et al., 2007). Landsat images courtesy of U.S. Geological Survey.

## 4.5 Warm environmental conditions during the mid-20[th] century trigger mass loss?

We hypothesise that anomalously high basal-melt rates in the mid-20[th] century could have acted as the trigger for the current mass loss of Totten Glacier. This could have manifested via anomalously warm conditions causing the ice shelf to detach from a pinning point. A candidate

could be the partial detachment from the main ice rumple near the grounding line, where the precise bed topography is still uncertain (Vaňková et al., 2023). At the very least, the period of no surface-undulation formation in the mid-20th century likely marked a period of unusually sustained higher basal-melt rates, with respect to conditions over the past 100 years (Fig. 8).

Validating the hypothesised period of anomalously warm conditions in the mid-20th century with reanalysis or other sources of observational data is challenging. Outputs from ice-ocean models reconstructing temporal variability in basal-melt rates back to the mid-20th century show conflicting signals (Gwyther et al., 2018; Kusahara et al., 2024), and the reliability of key climatic reanalysis input data throughout much of the early to mid-20th century is unknown. There also remains insufficient resolution of the sub-ice-shelf bathymetry that is crucial to simulating melt rates (Kusahara et al., 2024). Modern observations show synchronous ice-speed and ice-shelf thickness anomalies throughout the 2000s at major outlet glaciers across the wider Wilkes Land region (Miles et al., 2022). Therefore, any anomalously warm conditions in the mid-20th century could have been felt regionally and not just at Totten Glacier. We observe the collapse of a previously unreported ice shelf 50 km east of the ice front of Moscow University Ice Shelf between 1963 and 1973 (Fig. 10). The ice shelf is small (250 km$^2$) and did not buttress any large glaciers, so its collapse is inconsequential for sea-level rise. The unnamed ice shelf was previously adjoined to Voyeykov Ice Shelf through a mixture of fast ice and melange. Voyeykov Ice Shelf periodically collapses and regrows in response to changes in fast ice conditions (Arthur et al., 2021), but at the unnamed ice shelf we observe no signs of regrowth in the fifty years following its collapse (Fig. 10). The final stages of ice-shelf collapse may be preceded by years or even decades of gradual weakening (Walker et al., 2024), meaning the collapse of this unnamed ice shelf at some point between 1963 and 1973 is consistent with relatively warm environmental conditions in the years or decades earlier and is symbolic of environmental change in the mid-20th Century.

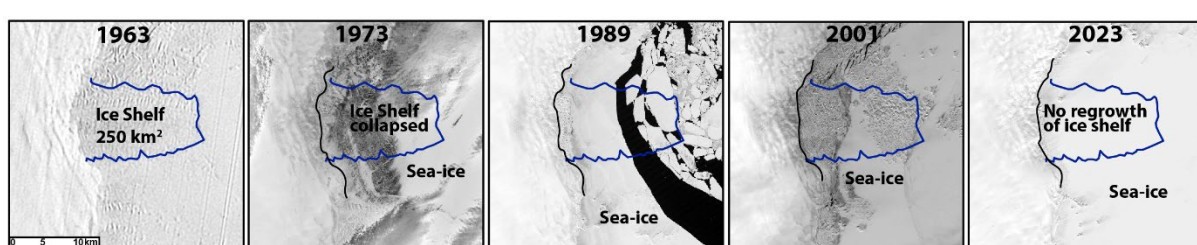

**Figure 10:** A series of satellite images showing the disintegration of an un-named ice shelf between 1963 and 1973, which has not regrown to a comparable extent in the 50 years since. The blue line denotes the boundary of the ice shelf in 1963. The black line is the ice-front position in 2023. Satellite images are courtesy of the U.S. Geological Survey. See Figure 1a for location.

The century-long record of surface undulations implies that the ~30-year record of satellite altimetry is not long enough to capture the full extent of the amplitude and wavelength of decadal variability in basal-melt anomalies. Numerical-modelling experiments have shown that forcing Totten Glacier with variable basal-melt rates, at similar amplitudes to those inferred by ~30 years of satellite-altimetry observations, has a net effect of reducing ice discharge and delaying grounding-line retreat in comparison to constant forcing (McCormack et al., 2021). Therefore, evidence of prolonged periods of higher sustained melt rates in the mid-20th century raises the possibility of similar episodes of accelerating retreat in the coming decades.

Elsewhere, interpretations of surface features on West Antarctica's Venable ice Shelf have revealed intermittent local re-grounding on a pinning point since the 1930s (Locke et al., 2025), implying similar decadal variability in basal-melt rates. A greater understanding of the processes and the magnitude of decadal variability is vital to narrowing uncertainties in future sea-level projections from Antarctica (e.g. Hanna et al., 2024). It could be that anomalous periods of warm oceanic conditions play a vital role in triggering cycles of glacier retreat. Perhaps analogous to this are Pine Island and Thwaites glaciers, where sediment records point to their retreat from stabilising pinning points in the 1940s (Clark et al., 2024; Smith et al., 2017) in response to anomalously warm environmental conditions (Steig et al., 2012). The key difference here is that bedrock geometry at Pine Island Glacier was conducive for marine ice-sheet instability and a rapid retreat (Reed et al., 2024), whereas the present geometry at Totten Glacier is not favourable for a similar rapid retreat in the near future (Morlighem et al., 2020).

## 5. Conclusion

Overall, our results have shown that the averaged decadal ice speed of Totten Ice Shelf has not accelerated over the past half-century. The combination of observations of inland thinning in the earliest satellite-altimetry records, grounding-line retreat between 1973 and 1989, and the absence of any long-term trends in snowfall, points towards the mass balance of Totten Glacier during this period being similar to present. Therefore, the initial increase in ice discharge that placed Totten Glacier out of balance must have taken place before 1973. The near century-long record of surface-undulation formation highlights an anomalous ~20-year period in the mid-20[th] century that we interpret as sustained higher-than-average basal-melt rates causing much more limited interaction between a bathymetric high point near the grounding line and the ice shelf. This interpretation is supported by satellite-altimetry observations that show waves of ice thickness variations travelling with ice flow that match the gap in surface-undulation formation. We hypothesise that these anomalously warm conditions in the mid-20[th] century could have accelerated mass loss of Totten Glacier by decoupling from a key pinning point, and/or marking a change in environmental conditions represented by more frequent and/or intense intrusions of mCDW onto the continental shelf. We also observe the collapse of a small neighbouring ice shelf during this period. Validating this proposed anomalously warm period is challenging because the drivers of interannual variability in melt rates at Totten Ice Shelf over the observational period are not fully understood. Perhaps more importantly, because surface undulations have been forming over the past ~30 years, our observations hint at warmer than present conditions in the recent past. Therefore, basal-melt rates calculated from short satellite-altimetry records that are often used as a baseline in numerical models to project into the future may not be capturing the full variability within the Totten glacier system.

**Data availability:** Landsat and ARGON imagery was provided free of charge by the U.S. Geological Survey Earth Resources Observation Science Center (https://earthexplorer.usgs.gov/, USGS, 2022). The MOA grounding line product is available at https://doi.org/10.7265/N5KP8037 (Haran, 2021). The ice shelf thickness change dataset from Smith et al. (2020) is available at http://hdl.handle.net/1773/45388 (last access: April 2022). The REMA DEM is available at https://doi.org/10.7910/DVN/X7NDNY (Howat et al., 2019). ICESat-2 data is available from https://nsidc.org/data/atl06/versions/6. The MEaSUREs Annual Antarctic Ice Velocity Maps are available at https://doi.org/10.5067/9T4EPQXTJYW9. Bathymetry and in situ basal-melt timeseries are available at https://doi.org/10.6084/m9.figshare.21824580 (Vaňková et al., 2023). Corrected Landsat imagery of Antarctic ice shelves from 1973 and 1989 is available at https://doi.org/10.7488/ds/3810 (Miles and Bingham, 2023). BedMachine version 3 is available at https://doi.org/10.5067/FPSU0V1MWUB6 (Morlighem, 2022). MEaSUREs ITS_LIVE Antarctic Grounded Ice Sheet Elevation Change is available at https://doi.org/10.5067/L3LSVDZS15ZV (Nilsson et al., 2025) Corrected ARGON imagery was provided by Ron Li. The decadal ice speed tracking vectors and grounding line mapping from this study are available at https://doi.org/10.7488/ds/7853.

**Acknowledgements:** Bertie W. J. Miles was supported by a Leverhulme Early Career Fellowship (ECF-2021-484). Tian Li is funded by the European Union's Horizon 2020 research and innovation programme through the project Arctic PASSION (grant number: 101003472) and the Leverhulme Early Career Fellowship (ECF-2024-157). Robert G. Bingham acknowledges funding from NERC (NE/S006613/1). We thank the editor Gong Cheng, and two anonymous referees for their constructive comments during the review process, which significantly improved the manuscript.

**Author Contribution:** B.M led the design of the study with input from T.L and R.G.B. T.L produced Figure 7d and carried out the ICESat-2 analysis. B.M led the manuscript writing with input from T.L and R.G.B

**Competing interests:** The authors declare that they have no conflict of interest.

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
