# Peer review of "Totten Ice Shelf history over the past century interpreted from satellite imagery"

_EGUsphere, 2024_

## Author Comment (AC1)

*Initial author response to reviewer comments*

*We would like to thank both reviewers for taking the time to provide valuable comments that will help improve this manuscript. We are pleased that both reviewers found our work interesting. Below, we provide initial responses to the reviewers' main points in the hope that we will be invited to submit a revision, allowing us to provide full responses and revise the manuscript accordingly. We note, we have also carefully considered all line and minor-level comments and are confident that we can address them all. However, to avoid unnecessary duplication, we do not respond to them individually in this document. We will, of course, include a full, traditional response to all comments upon submission of a revised manuscript.*

*Response to main comments*

*Both reviewers raised similar concerns surrounding the processes leading to the formation of the surface undulations and the reliability of our ice speed estimates as a proxy for grounding line flow. We address both concerns in a joint response below and have included the reviewers' original reports beneath our response.*

*1. How do surface undulations form?*

*In the revised version, we will include an additional figure to clarify that the surface undulations form due to time-varying re-grounding on a bathymetric high, driven by basal melt variability. This follows the helpful suggestions of both reviewers. **Specifically, we will add a panel showing the updated gravimetry-derived bathymetry of Totten from Vaňková et al. (2023).** The bathymetry near the ice rumple is complex. **The undulations form approximately 2–3 km away from the main rumple and immediately downstream of a secondary or connected bathymetric high.** As both reviewers acknowledged, bathymetry data beneath ice shelves is notoriously uncertain, and these features could easily be a few kilometers larger or smaller in extent and tens to hundreds of meters shallower or deeper. Nevertheless, we can demonstrate that the undulations are forming on or very close to a bathymetric high. A secondary argument is that no other plausible mechanism explains their formation. Reviewer 1 notes that these features cannot be basal crevasses—an argument we will incorporate into the revised text. Thus, the very presence of these undulations in satellite imagery provides strong evidence of a bathymetric high in the vicinity of their formation.*

*Reviewer 1 also asks whether we definitively conclude that the surface undulations are formed by basal melt variability, given the challenges in measuring melt rates. Fortunately, novel in situ observations of basal melt rates (2017–2019) from autonomous phase-sensitive radar exist at Totten (see Vaňková et al., 2023). The spatial location of this time series is very close to the surface undulation formation zone. **These observations show basal melt variability of 7–9 m $a^{-1}$, which is of a similar magnitude to the ~20 meters of surface elevation change associated with the surface undulations over two years. We will incorporate this into the revised manuscript and argue that this provides strong evidence that the undulations are formed by time-varying basal melt.** For reference, grounded ice thins at a long-term average rate (1992–2022) of ~1 m/a, though this rate varies over time within a range of approximately 0.25–1.75 m/a. Over multiple decades, long-term grounded ice thinning would result in progressively thinner ice flowing into the ice shelf, which likely contributes to the observed trend of progressively fewer and smaller surface undulations, as shown in Fig. 7c.*

*2. Regional ice speed as a proxy for flow speed at the grounding line*

*Reviewer 2 asks: "Can you show, using the MEASURES velocity record in the modern era, that velocity changes within this box are correlated with velocity changes across the grounding line in both relative*

*magnitude and phasing?" Similar concerns were raised by Reviewer 1, who asks: "Is velocity downstream of a pinning point actually representative of grounding line velocity?"*

*We have investigated this, and in the revised version, **we will include a new figure demonstrating that ice speed in the region where we track surface features is correlated with ice speed at the grounding line over the MEASURES era, both in terms of magnitude and phase**. This provides strong support for our initial hypothesis that ice speed 30 km downstream of the grounding line is a reasonable proxy for grounding line speed. Furthermore, if ice speed at the grounding line were not in phase with ice speed 30 km downstream over the long term, we would expect to see significant damage and crevassing. However, there is no evidence of any notable damage or crevassing in satellite imagery from the past 50 years.*

*Reviewer 2 also requested an expanded Figure 3 showing the feature we track over more epochs – we are very happy to provide this in the revised version, along with more detail on the uncertainties in tracking the feature.*

**References**

*Vaňková, I., Winberry, J. P., Cook, S., Nicholls, K. W., Greene, C. A., & Galton-Fenzi, B. K. (2023). High spatial melt rate variability near the Totten Glacier grounding zone explained by new bathymetry inversion. Geophysical Research Letters, 50, e2023GL102960. https://doi.org/10.1029/2023GL102960*

**Reviewer 1**

Miles et al. put forth a compelling study of ice rumple history on the Totten ice shelf over the past century. By analyzing the formation of repeated surface undulations that are likely due to interaction with seafloor topography, these authors suggest that a period in the absence of undulations is likely due to variation in basal melt rate. Additionally, they compute interannual and decadal velocities, both of which do not show a significant trend and have large anomalies. I think that after answering the following questions, the authors can greatly strengthen their case through analytical rigor and clarity. However, without these added changes, their arguments remain speculative and claims are overreaching.

**Major Points**

*1 . What actually are these undulations from?*

For the first half of the paper, I wasn't convinced that the undulations you show were not large basal crevasses/channels generated from flow past the pinning point. The main reason I thought of this is that there is the large (roughly stationary) pinning point that is several kilometers away from the location you analyze. I know you are aware of the phenomena of fractures propagating laterally beyond pinning points, as you have shown in Miles et al. 2024. However, the icesat2 time series that show surface growth flipped my opinion, as you would not expect surface raising of O(10m) from a basal crevasse. To not have other readers have this confusion, please put a map of bathymetry that explains how local topography may promote the observed undulations, rather than having them originate from the larger ice rumple. Additionally, mention that fractures can be generated downstream of pinning points, and suggest that the observed surface undulations are inconsistent with the surface signature of basal crevasses (e.g., Luckman et al., 2012; McGrath et al., 2012). I know bathymetry products like BedMachine are uncertain (and have associated uncertainty provided), and you cite some folks saying that, but I had to dig into bathymetry data to find that there is a tail or foot to the seamount inferred in that region that highly suggests that what you are looking at is formed locally due to regrounding. This will significantly strengthen your argument.

*2. Is velocity downstream of a pinning point actually representative of grounding line velocity?*

The observed velocity data is quite noisy - on long time scales there are fluctuations, and on short time scales there are similar or larger amplitude fluctuations (your Figure 5). I would expect this signature of noise in a stick-slip type system, which I would expect from the ungrounding, regrounding, and potentially fracturing that is generated by these seamounts. In a stick-slip system, I would not expect the velocity downstream of the stick-slip location to be similar to those upstream, aka the grounding line. Please provide a convincing explanation of why we would expect the velocities you measure to be representative of the grounding line, as I find lines 138-140 to be unconvincing and appear as too much extrapolation.

To help illustrate the point, the most extreme example of flow past a pinning point may be the Brunt ice shelf. If one tried to compute velocity downstream of the pinning point, then these would be dominated by the flowing/fracturing that occurs as ice moves past an obstacle, and the results may not be a sensible measure of the upstream or grounding line velocity. This is a more extreme case, where the buttressing provided by the pinning point significantly impacts the ice velocity both downstream and upstream of the obstacle. All in all, I am not convinced that the velocity measured is representative for the ice shelf as a whole, nor the grounding line, but mostly for the local region

in which it is measured. Please update your phrasing where appropriate - statements about whether a glacier is "in balance" is a measure of grounding line flux versus snowfall integrated from the ice divide, so I would remove or strongly minimize these large-scale claims.

*3. Can you definitively conclude that basal melt is the dominant signal by concluding that the changes in snowfall (thickening) and grounded ice thickness (H) and velocity (u, v) cannot correspond to the observed thickness variation? Currently, it is argued (e.g. line 18) but not sufficiently proven.*

You show in Figure 1a that you get about 2 meters a year of thinning of grounded ice over a half century window. I imagine this is an average rate. You show in Figure 8 that there is +-3 meters a year of melting on the ice shelf. Finally, you show 20 meters surface elevation changes in Figure 7a within 2 years. The mechanism you propose, largely basal melt variation, needs to get cyclic variations of melt pulses causing 20 meters of variation, as a baseline. I'm with you in that I think this is from the ocean, but mostly because I don't expect this to be from grounded ice thickness and velocity changes or from ice shelf snowfall. Given that basal melt rates may be challenging to measure, can you construct your argument in terms of ruling out the other two options?

The thin film mass balance equation is $\partial_t H + \text{div}(u\,H) = a - b$, as you are likely familiar with the terms: thickness tendency, divergence of ice flux with (depth-averaged) horizontal velocity vector $(u,v)$, and snowfall rate and melt rate on the opposite side of the equality. This is a more mathematical way of framing the above argument. Please incorporate this logic if possible to see if you can definitively conclude that the changes in $\partial_t H$ would be driven by $-b$ and not by $\text{div}(u\,H)$.

**Reviewer 2**

Miles et al. presents an interesting history of ice flow on the Totten Ice Shelf from the early 1970s to the present day, reconstructed from feature tracking in historical Landsat imagery. They also produce a history of ice surface undulations that they ascribe to varying degrees of interaction between the ice shelf and a hypothetical pinning point. They interpret this as a qualitative history of decadal variations in ice thickness that also suggests decadal variations in basal melt rates. Overall, they conclude that Totten Glacier was likely losing mass at a similar rate in 1970s, but that a period of high melting in the 1940s to 1960s may have initiated that mass loss.

Overall, the paper investigates an important question - the long-term history of mass loss in one of East Antarctica's most dynamic areas. However, the study cannot directly measure the variables of interest: flux across the grounding line and basal melt rates. What can be measured is the velocity of a single feature in the middle of the ice shelf and the presence or absence of surface undulations at certain locations. I think the paper's argument could be much stronger with more attention to giving quantitative, data-supported, or physics-based explanations that strongly link these proxy records to the variables of interest.

**Major Comments:**

**1.** More detail is needed on the manual feature tracking for velocity estimates. To what degree of precision can the exact same point on a feature be detected 10 years later? Is 1 pixel of error for the feature tracking a sufficient uncertainty bound given changes in features shape, illumination

conditions, etc over time? How reproducible is this tracking? (e.g., if someone else were to do the tracking with the same imagery, would they get the same numbers.)

A supplementary or appendix figure showing all the features that were tracked in each image pair and the specific points that were tracked would go a long way towards increasing confidence in these results. I'm imagining something like an expanded Figure 3.

**2.** I am not entirely convinced by the argument that velocity variations within the feature tracking box should reflect velocity variations across the grounding line. Since the feature tracking box is directly downstream of the ice rumple, it seems like it would be more indicative of flow as modulated by interactions with that pinning point, including fracture processes as the ice flows around or over a barrier. Can you show, using the MEASURES velocity record in the modern era, that velocity changes within this box are correlated with velocities changes across the grounding line in both relative magnitude and phasing?

**3.** Line 189 – this argument is not convincing to me. I generally understand how undulations might form as ice passes over a pinning point, but from Figure 1, it seems like these undulations are forming to the left side of the ice rumple, not directly down flow from it. Is there some submarine bedrock high or pinning point directly upstream of these undulations that is not marked on the maps or visible in the imagery? If not, what is the theory for undulations forming next to the pinning point? The 2D schematic in Figure 4 does not seem to capture the real geometry of the Totten Ice Shelf and would be more useful and convincing in 3D. I know that sub-ice shelf bathymetry is always highly uncertain, but if you are using BedMachine or some other project to infer the presence of a potential pinning point, it would greatly strengthen your argument to show that bathymetry.

On further digging in the literature, it seems that perhaps there is a lot of reliance on the conclusions from Roberts et al. (2017) when ascribing the origin of these undulations? If that is the basis for these conclusions, the paper would be strengthened by some review of their arguments for the reader who is not as informed about the history of these research on Totten.

---

## Author Response (AR1)

**Reviewer 1**

Miles et al. put forth a compelling study of ice rumple history on the Totten ice shelf over the past century. By analyzing the formation of repeated surface undulations that are likely due to interaction with seafloor topography, these authors suggest that a period in the absence of undulations is likely due to variation in basal melt rate. Additionally, they compute interannual and decadal velocities, both of which do not show a significant trend and have large anomalies. I think that after answering the following questions, the authors can greatly strengthen their case through analytical rigor and clarity. However, without these added changes, their arguments remain speculative and claims are overreaching.

*We are pleased the reviewer found our study compelling and we would like to take this opportunity to thank the reviewer for their very helpful suggestions to which we have responded in full below.*

**Major Points**

1. What actually are these undulations from?

    1. For the first half of the paper, I wasn't convinced that the undulations you show were not large basal crevasses/channels generated from flow past the pinning point. The main reason I thought of this is that there is the large (roughly stationary) pinning point that is several kilometers away from the location you analyze. I know you are aware of the phenomena of fractures propagating laterally beyond pinning points, as you have shown in Miles et al. 2024. However, the icesat2 time series that show surface growth flipped my opinion, as you would not expect surface raising of O(10m) from a basal crevasse. **To not have other readers have this confusion, please put a map of bathymetry that explains how local topography may promote the observed undulations, rather than having them originate from the larger ice rumple. Additionally, mention that fractures can be generated downstream of pinning points, and suggest that the observed surface undulations are inconsistent with the surface signature of basal crevasses (e.g., Luckman et al., 2012; McGrath et al., 2012).** I know bathymetry products like BedMachine are uncertain (and have associated uncertainty provided), and you cite some folks saying that, but I had to dig into bathymetry data to find that there is a tail or foot to the seamount inferred in that region that highly suggests that what you are looking at is formed locally due to regrounding. This will significantly strengthen your argument.

*This is a fair point and a very good suggestion. In the revised version we have included a new Figure 7 that incorporates two models of bathymetry for Totten Ice Shelf. This shows that the undulations are either forming on the corner of a secondary ice rumple slightly upstream of the main ice rumple (Vaňková et al., 2023), or directly underneath a bathymetric high point (i.e. the foot you mention above in BedMachine which is based on the Greenbaum et al. 2015 model). In the revised version we also include a new section 4.3 'Surface undulation formation' that explains why these undulations are caused by intermittent local re-grounding in response to time varying melt rates and not basal crevasses.*

2. Is velocity downstream of a pinning point actually representative of grounding line velocity?

1. The observed velocity data is quite noisy - on long time scales there are fluctuations, and on short time scales there are similar or larger amplitude fluctuations (your Figure 5). I would expect this signature of noise in a stick-slip type system, which I would expect from the ungrounding, regrounding, and potentially fracturing that is generated by these seamounts. **In a stick-slip system, I would not expect the velocity downstream of the stick-slip location to be similar to those upstream, aka the grounding line. Please provide a convincing explanation of why we would expect the velocities you measure to be representative of the grounding line, as I find lines 138-140 to be unconvincing and appear as too much extrapolation.**

   1. To help illustrate the point, the most extreme example of flow past a pinning point may be the Brunt ice shelf. If one tried to compute velocity downstream of the pinning point, then these would be dominated by the flowing/fracturing that occurs as ice moves past an obstacle, and the results may not be a sensible measure of the upstream or grounding line velocity. This is a more extreme case, where the buttressing provided by the pinning point significantly impacts the ice velocity both downstream and upstream of the obstacle. All in all, I am not convinced that the velocity measured is representative for the ice shelf as a whole, nor the grounding line, but mostly for the local region in which it is measured. Please update your phrasing where appropriate - statements about whether a glacier is "in balance" is a measure of grounding line flux versus snowfall integrated from the ice divide, so I would remove or strongly minimize these large-scale claims.

*In the revised version we have included a new Figure 5 that shows ice-speed anomalies derived from the MEaSUREs annual velocity mosaics for the grounding line and the feature tracking zone based on the suggestion from Reviewer 2. We only include years where there is full data coverage in both regions (2008-2009; 2010-2011 and 2013 onwards, i.e. from Landsat 8). This shows a correlation (P-value = 0.02) between the feature tracking zone and the grounding line. The fact that ice speed upstream of the pinning point at the grounding line is in phase with ice speed downstream of the pinning point provides convincing evidence that our initial assumption that ice speed within the feature tracking zone is a good representation of speed at the grounding line is correct.*

*We do not believe the example of the Brunt Ice Shelf is a fair comparison. At Brunt, ice is fully diverted around the pinning point leading to extensive fracturing. The Brunt pinning point is also located close to the ice front where the ice shelf is unconstrained. At Totten the pinning point is an ice rumple where ice flows over the top of the high and there is no visible fracturing in its immediate vicinity in any image over the past 50 years. Secondly, the rumple at Totten is close to, or possibly even an extension of the grounding line. These are not like for like comparisons.*

3. Can you definitively conclude that basal melt is the dominant signal by concluding that the changes in snowfall (thickening) and grounded ice thickness (H) and velocity (u, v) cannot correspond to the observed thickness variation? Currently, it is argued (e.g. line 18) but not sufficiently proven.

1. You show in Figure 1a that you get about 2 meters a year of thinning of grounded ice over a half century window. I imagine this is an average rate. You show in Figure 8 that there is +-3 meters a year of melting on the ice shelf. Finally, you show 20 meters surface elevation changes in Figure 7a within 2 years. The mechanism you propose, largely basal melt variation, needs to get cyclic variations of melt pulses causing 20 meters of variation, as a baseline. I'm with you in that I think this is from the ocean, but mostly because I don't expect this to be from grounded ice thickness and velocity changes or from ice shelf snowfall. Given that basal melt rates may be challenging to measure, can you construct your argument in terms of ruling out the other two options?

2. The thin film mass balance equation is partial_t H + div (u H) = a - b, as you are likely familiar with the terms: thickness tendency, divergence of ice flux with (depth-averaged) horizontal velocity vector (u,v), and snowfall rate and melt rate on the opposite side of the equality. This is a more mathematical way of framing the above argument. **Please incorporate this logic if possible to see if you can definitively conclude that the changes in partial_t H would be driven by -b and not by div (u H).**

*Basal melt rates are challenging to measure, but fortunately novel in situ basal-melt timeseries are available at Totten (Vaňková et al., 2023), and from just a few km upstream of where the surface undulations form. The timeseries is short (2 years) but shows basal melt variability of around 9 m $a^{-1}$, corresponding well with the 25 m elevation anomaly over 3 years of the surface undulations. We consider this alone to be convincing evidence that basal-melt variations are the dominant driver of surface-undulation formation. Changes in velocity near the zone of surface-undulation formation will ultimately be driven by changes in melt rates causing intermittent grounding, so this argument is somewhat circular. Variability associated with grounded ice thinning and surface mass balance is an order of magnitude lower at +/- 0.5 m $a^{-1}$ and +/- 0.2 m $a^{-1}$ respectively. We have added this argument into the new Section 3.3.1.*

**Minor Points**

1. Lines 16-17: Out of mass balance - insert the word mass.

*Amended*

2. Line 48: define inland thinning - I presume you mean the thinning of grounded ice, as in Figure 1a, but you don't define it

*Amended*

3. Lines 60-61: I would modify this sentence to say outlet glaciers lose mass through a flux imbalance, whereby ice mass loss aka discharge outweighs mass gain via snowfall and basal accretion under ice shelves.

*We have modified this sentence to: Outlet glaciers lose mass through a flux imbalance, whereby ice loss from ice discharge outweighs mass gain from surface mass balance processes.*

4. Lines 61-63: What is the point of this sentence? It seems unnecessary if you are focusing on building a story about Totten in EAIS.

   1. Lines 65-67: Part of the issue with the above sentence (61-63) is that it is used in a misleading manner, by tying mass loss in WAIS to a lack of acceleration of Totten Glacier in EAIS. You need to say in East Antarctica, since your last sentence is about West Antarctica and those conditions are different. CDW on the continental shelf does not imply the same environmental conditions.

*We have removed this sentence*

5. Lines 67-68: The reason that glaciers accelerate is due to a loss of buttressing. If you can't show that there has been a change in buttressing, then you won't get acceleration. Please revise your framing of the problem.

*Agreed. But we are interested in the changes in ice speed because there is no way to determine changes in buttressing going back to the 1970s because there are no data on ice-shelf geometry (except area). Changes in speed are an indicator of a change in buttressing.*

6. Line 69: Acceleration of the ASB or the ice shelf or grounding line retreat? Which acceleration are you focused on?

*Clarified as 'Totten glacier or ice shelf'*

7. Lines 70-71: Out of mass balance is a catch-all term, as technically glaciers are basically never actually balanced in an integrated sense given some environmental noise (Robel et al., 2018; Sergienko and Wingham, 2024; Sergienko and Haseloff, 2023). I think it is unreasonable to expect that glaciers would be in a steady state before the anthropocene in how you frame this statement.

*We agree that it unreasonable to expect glaciers to be in balance before the Anthropocene, but the wider point here is that the ice speed history of Totten is still not fully understood in the satellite era.*

*We have amended the sentence to state: "2) There was already a flux imbalance at the onset of the satellite era, meaning the Totten catchment was already loosing mass at the onset of satellite observations"*

8. Line 80: The surface elevation change is presumably an average rate. Can you say it is monotonic, or provide the reader with more relevant information for the thin film mass balance equation (see major comment 3)?

*Yes an average rate. In the revised version we have provided a timeseries of rate of grounded ice thinning between 1985 and 2020 in the supplement and discussed this in the results (Section 3.3.1).*

9. Lines 81-82: State that the rates of grounding line retreat are shown in red text in figure 1a.

   1. More Figure 1 aesthetics: Make the a and b labels more visible the font is too small, same from the text in the images.

*We have amended the figure caption as suggested and increased the label and text size in Figure 1.*

10. Figure 2
    1. Can you make the red circles wider or provide a border to them to help them stand out against the background?
    2. Subfigure c, phrase "formed through interaction with ice rumple" - please do much more to justify this and explain in the caption what you mean: local interaction with a seamount or lateral effects from the major surface elevation change that we can clearly visibly see to the right of the undulation train you are analyzing. I got quite lost thinking about how ice rumples can modify elevation patterns of nearby ice flow through a basal crevasse forming and/or basal channel melting to alter surface topography. See major point 1.

*We have made the red circles larger. We have removed the phrase from subfigure c. The processes driving the formation of surface undulations are also discussed in more detail in the new section 4.3.*

11. Section 2.2 Ice speed, first paragraph - great argument!

*Thanks*

12. Lines 135-138: either provide a hypothesis for what this feature is/how it formed, or state that it is of unknown formation. Your readers, who are scientists that enjoy problem-solving and oddities, may lose focus and try to figure it out for themselves.

*Clarified as unknown origin.*

13. Lines 138-141: I am unconvinced with this statement, see major point 2. I would appreciate more reasoning in your response rather than citing precedent.

*See response to major point 2.*

14. Lines 158-161: This seems interesting, but you don't include any figures/tables on this topic. Can you provide some more information via quantitative comparison, or remove this sentence?

*This sentence has been removed.*

15. Line 169: Is the channel along or perpendicular to the grounding line? This was unclear to me.

*I am afraid we are a little unclear what you mean here because you can argue both depending on your perspective. The channel is highlighted in Figure 1a.*

16. Lines 211-212: This statement added to my confusion around major point 1, as clearly fractures become a major feature of Totten ice shelf.

*This fracture field, 45 km downstream of the ice rumple, coincides with the ice shelf bending around Law Ice Dome and accelerating. We note there are no visible fractures near the ice rumple.*

17. Lines 212-214: See major point 3, please use the full mass balance equation, and mention that the ice thickness at the grounding line is changing due to grounded ice thinning and grounding line movement.

*See response to major point 3. We mention that ice thickness is changing due to grounding line movement and grounded ice thinning in the revised section 4.4.*

18. Figure 4: please improve this schematic, it is nearly identical to the simple schematic in Miles et al., 2024. Make it 3D, and provide surface and basal topography variations (currently you only have one). Could greatly strengthen your argument about cyclic melt cycles later on. Additionally, provide some CDW arrows or something to make the ocean distinct between warm periods and cooler periods, with grounding line melting. The ocean circulation, heat and stratification are key players in your argument.

*We appreciate this comment. We have tried experimenting making the schematic 3D, but could not make it work. We have improved the schematic on the new Figure 8 by tying it to each phase of surface-undulation formation, and including both grounded-ice thinning and grounding-line retreat. We would like to think that the incorporation of the bathymetry models and associated discussion, combined with the revised schematic, clarify how surface undulations are formed. The schematic does not show ocean circulation/stratification. Our main take-home message of the manuscript is that our interpretation of the undulations suggests high basal melt rates in the mid-20th century. What is driving multi-decadal variability in basal melt rates (as opposed to interannual variability) is unknown and beyond the scope of this manuscript.*

19. Lines 223-224: You should tie this point to the idea that there hasn't been a major loss of buttressing, and thus there isn't a major force change and subsequent ice acceleration change.

*We have incorporated this into the discussion in Section 4.1*

20. Lines 225-226: This sentence strengthens major point 2, and suggests that the decadal anomalies have such low signal-to-noise ratio (e.g. high frequency short term events) that your results for the decadal predictions might be a strong function of when the images were actually taken.

*We have addressed major point 2 above. Regarding the decadal ice-speed measurements also being a function of when the image was taken: yes, there is an element of this and is best visualised between 2000 and 2022 where we have regular decadal speed estimates and they follow a similar shape to the interannual anomalies. However, the important point here is that the anomalies are still centred around zero, hence presenting no evidence of a sustained acceleration of Totten over the past 50 years.*

21. Lines 240-242: Doesn't the gradual eastern channel widening through all of your time periods provide a different story compared to the hiatus in surface undulation

formation that you claim in the abstract and conclusion are indicative of warmer periods (more CDW entry) in the past? Please elaborate

*No. The first observation of the Eastern Channel is between 1973 and 1989. The proposed warm period and period of no surface undulation formation is earlier in time towards the mid-20$^{th}$ century.*

22. Figure 7

    1. Looking at the REMA elevation map with no axis units, labels, or flow orientation: I wish you had the star in every image, and noted whether the images are all properly aligned. It is hard to tell where the small inset is in a compared with b. Please label the flow direction if it's downwards in all images.

    2. Please align subfigure 1a with bathymetry data such as BedMachine. The ICESat-2 imagery is very convincing that this is not a basal channel or crevasse, and I would have appreciated seeing this much earlier in the paper. See major point 1.

    3. Subfigure b: fonts are too small

*We have removed this figure, and incorporated the ICESat-2 profiles into the new Figure 7. Bathymetry is also shown on the new Figure 7.*

23. Lines 290-293: Having a large difference in grounding line ice velocity compared with these downstream ice velocity measurements can be evidence for my major point 2.

*Please see response to major point 2*

24. Lines 321-326: You should do way more with these coastal oceanography studies that show roughly 3 and 7 year melt cycles. Please explain what causes the separation, or what those authors suggest. I think that if you did more with the data of those other studies, you would draw a larger audience from the oceanography community for citing your work.

*We have gone into more detail in the oceanographic processes driving the variability in melt rates in the new section 4.3*

25. Lines 332-334: either provide a citation or reference a figure where you can easily show this claim.

*This text has been re-written.*

26. Lines 361-363: Initial comment: Wasn't there an important ENSO in the 40s that was thought of as a trigger for Thwaites retreat? During paper review: You mention some papers later on that discuss this (lines 432-434), but you could bring the discussion or a reference of WAIS up to here.

*Yes there was, but there is no evidence that ENSO has the same effect on melt rate/accumulation variability at Totten in any of the modern records.*

27. Figure 8

    1. This melt rate is similar to the elevation change from thinning of grounded ice. Supports my arguments in major point 3.

    2. Why is the shelf getting thicker upstream? You are leaving the reader guessing/confused - for your argument, what really is important is grounding line thickness, and the melt rates downstream of the pinning point can't be that large as the features you track stay around for many years.

*Point 1: The thickness change in the ice shelf is not similar to elevation change from thinning of grounded ice. Grounded ice is thinning on average around 1 m $a^{-1}$. The ice-shelf thickness change shows elevation changes of -3 to + 3m $a^{-1}$ i.e. there is no positive/thickening of the grounded ice, it is more or less consistently getting thinner albeit at slightly different rates (see supplementary figure on grounded-ice thickness change). Only basal-melt variability through time can explain this pattern.*

*Point 2. We apologise for our argument being unclear here and we have re-written this section to clarify. The pattern of ice-shelf thickness change can be explained by the advection of ice of differing thicknesses downstream. The key point here is that over the 16-year time separation (2003-2019) of the ice-shelf thickness change dataset, ice will have been displaced by around 16 km (roughly 1 km year ice speed) – this broadly matches the lengths of the core thinning and thickening sections of the ice shelf (Fig. 9b). This revised paragraph is located in section 4.4.*

28. Lines 371-373: Another case of bathymetry being important, uncertain, but meanwhile there have been no maps of bathymetry in the entire paper, yet they are crucial to your surface undulation formation mechanism. This supports my major point 1. Bathymetry products often come with uncertainty - you can include this as well.

*We agree, and have included bathymetry maps in the revised manuscript.*

Lines 403-405: Why not make more of a point of mentioning this as an aside in your abstract and/or conclusion? This is good to know and is fully tucked away deep in a paper that doesn't highlight it.

*We have added a line on this in the abstract and conclusion.*

29. Figure 9 - How are you drawing that ice shelf boundary? Is it land-fast sea ice, or glacial ice, or a mix?

*Ice-shelf boundary – we have clarified this in the figure caption.*

30. Lines 444-445: see minor points 1, 3, and 7 for the same idea - I think it is unreasonable to assume that a glacier is ever truly "in balance" with respect to grounding line flux, which will oscillate the observed mass loss (and SLR for marine-terminating glaciers).

*We have amended this sentence to state 'similar to modern-day mass balance'*

31. Lines 450-454: same idea as line above, and accelerating the detachment is very confusing language. Just say decoupling or losing contact.

*Amended to 'could have accelerated mass loss of Totten Glacier by decoupling from a key pinning point'*

**References**

1. Luckman, A., Jansen, D., Kulessa, B., King, E. C., Sammonds, P., and Benn, D. I.: Basal crevasses in Larsen C Ice Shelf and implications for their global abundance, The Cryosphere, 6, 113–123, https://doi.org/10.5194/tc-6-113-2012, 2012.

2. McGrath, Daniel, Konrad Steffen, Ted Scambos, Harihar Rajaram, Gino Casassa, and Jose Luis Rodriguez Lagos. 2012. "Basal Crevasses and Associated Surface Crevassing on the Larsen C Ice Shelf, Antarctica, and Their Role in Ice-Shelf Instability." Annals of Glaciology 53 (60): 10–18. https://doi.org/10.3189/2012AoG60A005.

3. Robel, Alexander A., Gerard H. Roe, and Marianne Haseloff. 2018. "Response of Marine-Terminating Glaciers to Forcing: Time Scales, Sensitivities, Instabilities, and Stochastic Dynamics." *Journal of Geophysical Research: Earth Surface* 123 (9): 2205–27. https://doi.org/10.1029/2018JF004709.

4. Sergienko, Olga, and Duncan John Wingham. 2024. "Diverse Behaviors of Marine Ice Sheets in Response to Temporal Variability of the Atmospheric and Basal Conditions." *Journal of Glaciology* 70 (January):e52. https://doi.org/10.1017/jog.2024.43.

5. Sergienko, Olga, and Marianne Haseloff. 2023. "'Stable' and 'Unstable' Are Not Useful Descriptions of Marine Ice Sheets in the Earth's Climate System." *Journal of Glaciology* 69 (277): 1483–99. https://doi.org/10.1017/jog.2023.40.

6. Miles, B.W.J., Bingham, R.G. Progressive unanchoring of Antarctic ice shelves since 1973. Nature 626, 785–791 (2024). https://doi.org/10.1038/s41586-024-07049-0

**Reviewer 2**

Miles et al. presents an interesting history of ice flow on the Totten Ice Shelf from the early 1970s to the present day, reconstructed from feature tracking in historical Landsat imagery. They also produce a history of ice surface undulations that they ascribe to varying degrees of interaction between the ice shelf and a hypothetical pinning point. They interpret this as a qualitative history of decadal variations in ice thickness that also suggests decadal variations in basal melt rates. Overall, they conclude that Totten Glacier was likely losing mass at a similar rate in 1970s, but that a period of high melting in the 1940s to 1960s may have initiated that mass loss.

Overall, the paper investigates an important question - the long-term history of mass loss in one of East Antarctica's most dynamic areas. However, the study cannot directly measure the variables of interest: flux across the grounding line and basal melt rates. What can be measured is the velocity of a single feature in the middle of the ice shelf and the presence or absence of surface undulations at certain locations. I think the paper's argument could be much stronger with more attention to giving quantitative, data-supported, or physics-based explanations that strongly link these proxy records to the variables of interest.

*We are pleased the reviewer found our study interesting and we would like to take this opportunity to thank the reviewer for their very helpful suggestions to which we have responded below.*

**Major Comments:**

**[1]** More detail is needed on the manual feature tracking for velocity estimates. To what degree of precision can the exact same point on a feature be detected 10 years later? Is 1 pixel of error for the feature tracking a sufficient uncertainty bound given changes in features shape, illumination conditions, etc over time? How reproducible is this tracking? (e.g., if someone else were to do the tracking with the same imagery, would they get the same numbers.)

A supplementary or appendix figure showing all the features that were tracked in each image pair and the specific points that were tracked would go a long way towards increasing confidence in these results. I'm imagining something like an expanded Figure 3.

*Some surface features can change shape over the course of days, others can persist unchanged for decades. Therefore, the time separation between images is not the determining factor of feature-tracking accuracy i.e. you could have two images separated by just a month, but the features could deteriorate. Rather, the principal determinant for our analysis was selecting features that persistently keep their shape. We tried to show that a feature keeping its shape exceptionally well over long periods in Figure 3, which in theory is the poorest quality image pair with the lowest spatial resolution and separated over the longest time period (16 years), yet the feature is still clearly visible in both images. In the revised version we include a supplementary figure of the other more modern image pairs as requested. We are confident that +/- 1 pixel is a reasonable uncertainty because we can clearly see that the feature maintains its shape when we have the annual imagery from 2000 onwards. We also note that all feature-tracking vectors are available to download for others to inspect as mentioned in the data availability section. We reiterate here that tracking features over longer timescales and*

*therefore longer distances actually significantly reduces uncertainty. For example, if we were to assume a pixel uncertainty of +/- 4 pixels – which is unrealistically large because nine pixels (the true pixel plus four either side) would be longer than the entire feature – the error would only increase to +/- 1.8%. Tracking features over shorter timescales, even with modern imagery, has inherently larger errors.*

**[2]** I am not entirely convinced by the argument that velocity variations within the feature tracking box should reflect velocity variations across the grounding line. Since the feature tracking box is directly downstream of the ice rumple, it seems like it would be more indicative of flow as modulated by interactions with that pinning point, including fracture processes as the ice flows around or over a barrier. Can you show, using the MEASURES velocity record in the modern era, that velocity changes within this box are correlated with velocities changes across the grounding line in both relative magnitude and phasing?

*In the revised version we have included a new Figure 5 that shows ice-speed anomalies derived from the MEaSUREs annual-velocity mosaics for the grounding line and the feature-tracking zone based on your helpful suggestion. We only include years where there is full data coverage in both regions (2008-2009; 2010-2011 and 2013 onwards, i.e. from Landsat 8 era). This shows a correlation (P-value = 0.02) between the feature-tracking zone and the grounding line. The fact that ice speed upstream of the pinning point at the grounding line is in phase with ice speed downstream of the pinning point provides convincing evidence that our initial assumption that ice speed within the feature tracking zone is a good representation of speed at the grounding line. We note that there is no visible evidence of any fractures within the immediate vicinity of the ice rumple in any satellite imagery.*

**[3]** Line 189 – this argument is not convincing to me. I generally understand how undulations might form as ice passes over a pinning point, but from Figure 1, it seems like these undulations are forming to the left side of the ice rumple, not directly down flow from it. Is there some submarine bedrock high or pinning point directly upstream of these undulations that is not marked on the maps or visible in the imagery? If not, what is the theory for undulations forming next to the pinning point? The 2D schematic in Figure 4 does not seem to capture the real geometry of the Totten Ice Shelf and would be more useful and convincing in 3D. I know that sub-ice shelf bathymetry is always highly uncertain, but if you are using BedMachine or some other project to infer the presence of a potential pinning point, it would greatly strengthen your argument to show that bathymetry.

*This is a very good suggestion. In the revised version we have included a new Figure 7 that incorporates two models of bathymetry for Totten Ice Shelf. This shows that the undulations are either forming on the corner of a secondary ice rumple slightly upstream of the main ice rumple (Vaňková et al., 2023), or directly underneath a bathymetric high point in the case of the Greenbaum et al. (2015) solution (used in BedMachine), depending on the accuracy of the bathymetry product. In the revised version we also include a new Section 4.3 'Surface-undulation formation' that explains why these undulations are caused by intermittent local re-grounding in response to time varying melt rates.*

*We have tried experimenting to make the schematic 3D, but could not make it work. We have improved the schematic in the new Figure 8 by tying it to each phase of surface-undulation formation, and including both grounded-ice thinning and grounding-line retreat. We would like to think that the incorporation of the bathymetry models and associated discussion, combined with the revised schematic, clarify how surface undulations are formed.*

On further digging in the literature, it seems that perhaps there is a lot of reliance on the conclusions from Roberts et al. (2017) when ascribing the origin of these undulations? If that is the basis for these conclusions, the paper would be strengthened by some review of their arguments for the reader who is not as informed about the history of these research on Totten.

*This study is indeed influenced by the Roberts et al. (2017) that first discussed/observed the surface undulations, as cited throughout the manuscript. We have added a brief summary of the Roberts hypothesis regarding the surface undulations in section 2.4. However, in the revised version we go further than the Roberts study in terms of showing how these undulation form – largely by incorporating a number of new datasets in Section 4.3 (two bathymetry models, ICESat2, in situ basal-melt observations) that were not available in 2017.*

**Minor Comments:**

Line 34 – "…detachment of **the** ice shelf from pinning points…" Which ice shelf is being discussed here? This is confusing since the sentence seems to refer to the entire ASB sector.

*Amended to 'ice shelves from pinning points'*

Line 60 – surface mass balance would be better than snowfall here. I suppose on Totten that SMB is dominated by snowfall, but since the sentence is general one about outlet glaciers, I would recommend using SMB.

*Amended to surface mass balance.*

Line 64 – I don't see where Kim et al. (2024) ascribes the mass loss they measure to the thinning of ice shelves. Ice shelves are barely mentioned in the paper, which to my reading it is more focused on measuring the partitioning between SMB and discharge.

*This sentence has been removed.*

Figure 1 – the label text is quite small, and the images are low dpi which makes the labels hard to read even when zoomed in on the pdf. Ground line retreat rates are given as distance, not rates. The timescale needs to be specified. If the black line in panel b is the MODIS 2009 grounding line, why are there multiple discontinuous black lines in many places? It looks like multiple grounding line estimates are overlaid?

*We have increased the font size and DPI of the image. We have provided the timescale of the grounding line migrations. We have corrected the figure caption to state that the grounding lines shown in Figure 1b cover the period 2017-2019. We think it is important to show both the MODIS grounding line (break in slope) which has the advantage that it is spatially complete, and the grounding lines/zone derived by InSAR.*

Figure 2 – the purple and green "feature tracking vectors" need more explanation. What are they and what do the different colors represent? What should the reader take from their presence on the figure?

*We have removed the feature tracking vectors from the image.*

Line 159 – this difference between MEASURES and ITS_LIVE seems significant. Do your results compare favorably with one or the other, or present a third interpretation? Why are there such stark differences between the products?

*This sentence has been removed.*

Line 210 – would be very helpful to have this flowline on a map somewhere. In general, having flow vectors for velocity on one of the maps would help a lot with interpreting much of the discussion on how features are related to one another along the shelf.

*Good suggestion, we have included a flowline on the new Figure 7.*

Figure 6 – this figure needs a locator map showing the whole ice shelf and where these individual sections fall.

*We have added to the Figure 6 caption that the location of the panels is shown in Figure 1b (white boxes).*

Line 288 – this seems to counteract your argument that the velocity anomalies within your feature tracking area should be indicative of the velocity across the grounding line.

*Please see response to major point 2.*

Line 321 – this is so key to your argument, it could use more development. Can you discuss more the drivers of these melt cycles, whether they apply over the whole ice shelf, how they would impact ice thickness at the grounding line, etc? What are typical basal melt rates on the Totten Ice Shelf from this study and magnitude of the change over the cycle? Could these variations in that melt rate reasonably explain the 25 m variations in surface elevation? Is it possible to directly align these modeled melt cycles with the undulation wavelengths like in Figure

*We have gone into more detail into the processes driving variability in the new Section 4.3, specifically showing that the magnitude of variability in basal melt rate broadly matches the 25 m variations in surface elevation over 3 years*

Line 376 – this seems somewhat speculative given that a clear physical mechanism for the formation of these undulations to the side of the ice rumple has not been clearly presented.

*We have provided a clear mechanism in the new Section 4.3.*

Line 389 – What is the decadal variability in melt rates? Does it align with the scale of melt rates that would be needed to observe or not observe the formation of these undulations?

*This is also addressed in the new section 4.3.*

---

## Referee Report (RR1)

Dear Editor,

Miles and others have done a wonderful job strengthening the paper and addressing reviewer questions after the latest round of reviews. I only have minor comments.

**Minor comments**

1. Grounding line evolution without yearly labels in Figures 1b, 5, and 7abc. Please add colors and text to describe which line corresponds to which year.

2. Lines 202-207 in the difference PDF (called diff henceforth), 175-178 in the new main PDF (called main henceforth), please split into two sentences.

3. Line 254 in diff, 226 in main: remove the word subtle, as the differences are fairly pronounced.

4. Line 277 in diff: Do you mean to say ice speed anomalies rather than ice speed? Figure 5 does not compare speed magnitude quantitatively (we get a qualitative sense in a) but rather measure speed anomalies in b. Similarly, consider changing appropriate to reasonable for proxies, with my point expanded below.

   a. More on this point: In Figure 5b, before 2014, the two locations overlap in error, but the sign isn't the same. In 2014, the velocities are beyond uncertainty in percent speed change. Afterward, yes, there is more similarity in phase, with large uncertainties. I think that today this is a reasonable proxy for the regions of interest, but still would not exactly expect this to always hold. I am glad that the

previous lines 138-140 in the previous main document, 148-153 in the main, have been removed.

5. Line 308 in diff: Add a space before the first parenthesis at (location …).

6. Section 3.3.1: Consider concluding this paragraph by claiming that the dominant balance in the mass conservation equation is between thinning and basal melting.

7. Great additions to Figure 7, I found it quite captivating.

8. Lines 414-415 in diff: Please consider putting something, such as a 1 and a 2 with arrows perhaps, for the main and secondary rumples this sentence references. Additionally, I'd reference the animation S1 in this paragraph, it is quite informative to see.

9. Lines 432-433 in diff: I do not find this to be a strong claim in the current phrasing, although I agree with the larger point of undersampling. I would consider rephrasing to follow the idea that the full range of variability of basal melt rates may not have been captured in the short two-year window, which is further supported by oceanographic studies of CDW variability on longer timescales, etc.

10. Figure S2: Why is the y-axis of this plot in m yr^-2? Presumably a typo?

11. Paragraph 1 of section 4.3, lines 463-480 in diff: You should break this up into two paragraphs. First, end the first paragraph before "A key implication …". Second, write about your considerations of three non-exclusive hypotheses that may permit smaller undulations in phase 3 than in phase 1.

12. Lines 519-21 in diff: Please refer to your Figure 9b advection hypothesis as Eulerian advection.

    a. Broader point: I think the argument (explaining thickness change in Figure 9b with Eulerian advection) would be more convincing if you explain what one

would expect with high melt rates in phase 2 at the GL, followed by a hiatus (or intermittent high and low melt rates in Phase 3). Then, by drawing your readers on board with expectations, showing both the distance and phase boundary are in line with the observations may enhance the delivery of the claim.

13. Figure 9a legend title: Basal (typo).

---

## Author Response (AR2)

*We would like to thank the editor and both reviewers for their comments which have helped further strengthen this manuscript.*

**Editor**

Thank you for thoroughly addressing the reviewers' comments. The revised manuscript has a significant improvement, including higher-quality figures and expanded discussion and supplementary information. Both reviewers are satisfied with the changes. Nonetheless, I would like to raise a few minor but important concerns that should be addressed prior to final publication:

1. As noted by Reviewer #1, the definition of "1-pixel uncertainty" remains somewhat arbitrary. I recommend the authors clarify this term explicitly in the text to reflect its empirical basis, and consider the uncertainty estimation approach proposed by the reviewer.

*We appreciate the argument made by Reviewer 1 that there may be subtle changes in the shape feature through time. However, we do not believe their proposed solution addresses this issue. If the feature changes shape through time, then having multiple authors track the feature still doesn't work. Secondly, there is strong argument that author skill levels and experience in working with the older imagery and tracking features are important here. An author with no or limited experience is not directly comparable to an author with extensive experience.*

*Out of caution we have increased the uncertainty associated with feature tracking over our decadal image pairs to two pixels, giving a total uncertainty of ±3 pixels. We note this is a very conservative estimate of uncertainty compared to other published studies.*

2. In Figure 5c, the red curve from 2008 to 2020 appears inconsistent with the light blue curve in Fig. 5b. As a sanity check, the authors should calculate the velocity anomaly from MEaSUREs for the same location of the undulation and include it in Fig. 5c for comparison.

*Yes, there is some discrepancy between the MEaSUREs data in the feature tracking zone (light blue line) and manual tracking of the feature over biennial timescales (red line) in Figure 5. This is to be expected because they are not over the same timescales. Our manually derived estimates are over ~2 years depending on image availability. The annual MEaSUREs data are from some point in the 12-month period between 1st July and 30th June in the given time period e.g. 1st July 2013- 30th June 2014. There is no way of knowing whether the data in these mosaics truly represent average flow speeds across the entire time period, or alternately come from only certain months/weeks within the 12 month time period. Given that ice speed anomalies at Totten can rapidly change, we do not believe there is a way to meaningfully compare ice speeds over different timescales. We note the general pattern of a peak in ~2007-2010 (+4%) and trough in ~2013-2015 (-4%) is consistent with Greene et al., 2017 who used a dataset independent to MEaSUREs.*

*See Figure 2 in Greene et al. ,Wind causes Totten Ice Shelf melt and acceleration. Sci. Adv.3, e1701681 (2017). DOI:10.1126/sciadv.1701681*

3. The manuscript conflates the concepts of ice velocity, ice discharge, and mass loss in several sections (abstract, Section 4.1, lines 326–327, and conclusion). While the authors convincingly show no acceleration in velocity over the past five decades, however, ice discharge also depends on grounding-line ice thickness, for which no direct evidence of constancy is presented. Given the sensitivity of mass balance to small changes in surface mass balance and discharge, more

quantitative evidence need to be shown before definitive conclusions are drawn. The authors should revise these discussions for greater conceptual clarity and precision.

*We have improved section 4.1 by confirming that there have been no major changes in the thickness of ice over the grounding line. The slope along which the grounding line is migrating is shallow at -0.3 degrees, so the published estimate of 3 km grounding-line retreat between 1996 and 2013 accounts for a 15 m deepening in bed elevation. Grounded ice thinning is approximately 0.5 m / year, so 25 m over 50 years. Given that Ice flowing over the grounding line is 2300 m thick, so any changes in ice thickness are proportionally negligible in the context of Totten's current ~10% imbalance.*

4. Regarding the concern from Reviewer #1 on basal melt rates: there is insufficient evidence to assert that basal melt must equal surface thinning. The surface elevation change of ~9 m implies a total thickness change of ~81 m under flotation assumptions. Ice dynamics must be considered in this context, and the discussion should clearly reflect this complexity.

*We have amended the text to highlight the role of ice dynamics in surface undulation formation.*

**Minor:**

- Line 160: Please ensure Tables 1 and 2 are reformatted such that dates in the first two columns occupy only one row each. Additionally, avoid splitting column headers like "Days", "Distance", and "Uncertainty" into separate rows.

*Amended*

Once these issues are addressed, the manuscript should be suitable for publication in The Cryosphere.

Best regards,

Cheng Gong

**Reviewer 1**

Dear Editor,

Miles and others have done a wonderful job strengthening the paper and addressing reviewer questions after the latest round of reviews. I only have minor comments.

Minor comments

1. Grounding line evolution without yearly labels in Figures 1b, 5, and 7abc. Please add colors and text to describe which line corresponds to which year.

*For simplicity we now only show one grounding line from October 2017 which was the most spatially complete of the grounding lines between 2017 and 2019 that we originally showed.*

2. Lines 202-207 in the difference PDF (called diff henceforth), 175-178 in the new main PDF (called main henceforth), please split into two sentences.

*We have split the sentence into two.*

3. Line 254 in diff, 226 in main: remove the word subtle, as the differences are fairly pronounced.

*Removed.*

4. Line 277 in diff: Do you mean to say ice speed anomalies rather than ice speed? Figure 5 does not compare speed magnitude quantitatively (we get a qualitative sense in a) but rather measure speed anomalies in b. Similarly, consider changing appropriate to reasonable for proxies, with my point expanded below.

a. More on this point: In Figure 5b, before 2014, the two locations overlap in error, but the sign isn't the same. In 2014, the velocities are beyond uncertainty in percent speed change. Afterward, yes, there is more similarity in phase, with large uncertainties. I think that today this is a reasonable proxy for the regions of interest, but still would not exactly expect this to always hold. I am glad that the previous lines 138-140 in the previous main document, 148-153 in the main, have been removed.

*Thanks for pointing this out, we did mean to say 'ice speed anomalies' and we have amended this. We have also changed 'appropriate' to 'reasonable'.*

5. Line 308 in diff: Add a space before the first parenthesis at (location …).

*Space added.*

6. Section 3.3.1: Consider concluding this paragraph by claiming that the dominant balance in the mass conservation equation is between thinning and basal melting.

*We do not introduce the equation at any other point of the manuscript, so we do not think it would be well placed here.*

7. Great additions to Figure 7, I found it quite captivating.

*Thanks*

8. Lines 414-415 in diff: Please consider putting something, such as a 1 and a 2 with arrows perhaps, for the main and secondary rumples this sentence references. Additionally, I'd reference the animation S1 in this paragraph, it is quite informative to see.

*We have added the arrows and referenced animation S1.*

9. Lines 432-433 in diff: I do not find this to be a strong claim in the current phrasing, although I agree with the larger point of undersampling. I would consider rephrasing to follow the idea that the full range of variability of basal melt rates may not have been captured in the short two-year window, which is further supported by oceanographic studies of CDW variability on longer timescales, etc.

*We have amended the text to address this point.*

10. Figure S2: Why is the y-axis of this plot in m yr^-2? Presumably a typo?

*Amended*

11. Paragraph 1 of section 4.3, lines 463-480 in diff: You should break this up into two paragraphs. First, end the first paragraph before "A key implication …". Second, write about your considerations of three non-exclusive hypotheses that may permit smaller undulations in phase 3 than in phase 1.

*We have decided to keep the paragraph structure as is because the paragraph is all on the same subject.*

12. Lines 519-21 in diff: Please refer to your Figure 9b advection hypothesis as Eulerian advection.

a. Broader point: I think the argument (explaining thickness change in Figure 9b with Eulerian advection) would be more convincing if you explain what one would expect with high melt rates in phase 2 at the GL, followed by a hiatus (or intermittent high and low melt rates in Phase 3). Then, by drawing your readers on board with expectations, showing both the distance and phase boundary are in line with the observations may enhance the delivery of the claim.

*We have referred to Eulerian advection and have re-ordered the text as suggested*

13. Figure 9a legend title: Basal (typo).

*Amended*

**Reviewer 2**

I think the paper has substantially improved based on these latest revisions. I am satisfied with the responses to all my minor comments. In my original review, I had three major comments on the manuscript:

[1] I was uncertain whether a 1-pixel uncertainty for manual feature tracking is sufficient/appropriate given the long time period between images.

I really appreciate the addition of Figure S1 showing the feature tracking vectors and the stability of the surface feature used for the velocity estimates. I totally agree with the authors reasoning that using decade long offsets between images reduces the overall uncertainty in velocity due to the large feature displacements relative to the picking uncertainty. However, I still do not see how it is possible to pick the exact same spot on this feature to within only 1 pixel error when the precise shape of the feature is shifting over time, as shown in Figure S1. For example, looking at the 2007 vs. 2017 images, the feature in question seems to have evolved from a single ridge or peak to a double ridge with a trough in the middle. My argument is that uncertainty could be better quantified by having 2-3 people independently pick these features in the imagery and then comparing their displacement estimates, rather than assuming an error of 1 pixel based on past literature. Given that the study only uses ~17 images, I do think this is a manageable amount of effort for an accurate quantification of uncertainty.

*We have addressed this comment in the response to the editor.*

[2] I felt more evidence was needed that the velocities within the feature tracking box are a good proxy for velocities across the grounding line.

The authors addressed this question very comprehensively in their new Figure 5b and I am satisfied that the feature tracking regions is a reasonable proxy for overall trend in velocity across the grounding line.

*Thanks.*

[3] I felt a more thorough discussion of how the surface undulations form was warranted.

The new Section 4.3 and Figure 7 largely resolve my concerns. I have just one comment on the discussion of the magnitude of the basal melt rates:

At lines 377 to 383, the authors argue that 25 m of differential surface-elevation anomalies over the course of three years requires basal melt rates of a similar size. I interpret this to mean that the authors assume all the changes in surface elevation (e.g. ice thickness) across these undulations come from time-varying basal mass balance. But my understanding is also that the authors are arguing that the undulations are forming because of grounding/ungrounding from a basal pinning point. If that is the case, I would expect the surface elevation change to also come from changes in the local divergence of the ice flux. When the ice is grounded, local compression due to the sudden spatial shift in basal traction could lead to a local pile-up of ice, and when it is ungrounded this doesn't occur since the shelf is locally in extension. In that case, the relevant magnitude of basal melting might just be whatever is required to ground/unground the shelf, which could be quite a bit more or less than the observed elevation change due to the ice dynamics and imprint of the basal topography. This argument would be stronger if the authors could clarify their conception of the role of basal melting vs. local changes in the velocity gradients at the pinning point in creating these undulations and explain why they conclude they think that ice shelf thinning from basal melting dominates the actual elevation signature (not just the existence of the undulations or their spacing).

(I realize there is inherently some circularity here, since the change in basal melt rates causes the change the velocity gradients through grounding/ungrounding. I agree that basal melt rates have to play an important role here. I'm just struggling to understand why, quantitatively, the surface elevation change rate you measure has to be in directly equal to the basal melt rate and would not also be majorly impacted by the change in ice dynamics due to the basal melting.)

*We have addressed this comment in the response to the editor.*